# The zinc-finger transcription factor Blimp1/Prdm1 is required for uterine remodelling and repair in the mouse

Maria-Eleni Xypolita[1], Mubeen Goolam [1,2], Elizabeth K. Bikoff[1], Elizabeth J. Robertson [1] ✉ & Arne W. Mould [1,3]

The zinc finger transcription factor Blimp1/PRDM1 regulates gene expression in diverse cell types. Its activity controls the maternal decidual response at early post-implantation stages of development. The present experiments demonstrate surprisingly that Blimp1 activity in the uterus is required for tissue remodelling at sites of embryonic failure. Moreover Blimp1 mutant females are refractory to RU486 induced decidual shedding. RNA-seq together with immunostaining experiments strongly suggest that the failure to up-regulate expression of the matrix metalloprotease Mmp10 in combination with insufficient suppression of BMP signalling, likely explain Blimp1-dependent phenotypic changes. In the post-partum uterus Blimp1 together with Mmp10 are highly upregulated at sites of tissue repair following placental detachment. Conditional Blimp1 removal significantly impairs the re-epithelization process and severely impacts involution of the endometrium and luminal epithelium. Overall these results identify Blimp1 as a master regulator of uterine tissue remodelling and repair.

The uterus normally undergoes continuous rounds of tissue remodelling during the oestrus cycle in response to fluctuating levels of ovarian hormones to ensure the uterine environment is receptive for embryo implantation[1,2]. Local disruption of the luminal epithelial layer and rapid stromal cell decidualization during implantation results in envelopment of the embryo by maternal tissue[3]. Subsequent interactions between the invading fetal trophoblasts and the maternal decidual cells promote vascular remodelling to connect the developing fetal-placental unit with the maternal circulation. After parturition, in response to falling levels of progesterone, the uterus subsequently undergoes a process of involution to remodel the luminal epithelium and regenerate lost and damaged tissues[4,5]. Following detachment and expulsion of the placenta, localized tissue regeneration is required to restore the integrity of the luminal epithelial layer[4,5]. Remarkably female mice immediately re-enter oestrus and mate following birth of their offspring. Via involution, the uterus rapidly acquires a receptive

state within 3-4 days post-partum. Additionally throughout gestation, the mouse uterus has the ability to respond to failure of abnormal embryos. This poorly understood process, termed "resorption", results in focal clearance of embryonic remnants and associated maternal tissues. Localized tissue remodelling leads to regeneration and repair of the stroma and luminal epithelial layer.

We recently described essential contributions made at early stages of pregnancy by the zinc-finger SET domain transcription factor Blimp1. Blimp1 mutant embryos fail at 10.5 days post coitum (dpc) due to defective placental morphogenesis[6,7]. We showed that Blimp1 expression in the embryonic trophoblast lineage is required for specification of the highly specialized spiral artery trophoblast giant cells that mediate invasion of the maternal deciduum and remodelling of the maternal blood vessels[8]. Expression is locally induced at implantation sites in the uterine luminal epithelium and then robustly up-regulated in the primary decidual zone (PDZ) immediately

[1]Sir William Dunn School of Pathology, University of Oxford, South Parks Road, Oxford OX1 3RE, UK. [2]Department of Human Biology and Neuroscience Institute, University of Cape Town, Cape Town 7925, South Africa. [3]Department of Psychiatry, University of Oxford, Warneford Hospital, Oxford OX3 7JX, UK. ✉ e-mail: elizabeth.robertson@path.ox.ac.uk

surrounding the embryo[9]. Conditional inactivation of Blimp1 activity in the uterus using the well described progesterone receptor Cre (PRCre) deleter strain[10] does not disturb stromal decidualization but rather results in defective formation of the PDZ barrier and abnormal trophectoderm invasion causing embryonic lethality at day 6.5[9]. Surprisingly here we discovered in PrCre Blimp1 mutant females that the residual decidual swellings, while devoid of viable embryos, fail to be reabsorbed during gestation. The uterine tissue is unable to remodel and remarkably at the equivalent of up to four weeks post-partum (PP) decidual structures formed at implantation persist and occlude the uterine horns.

To further investigate Blimp1 functional contributions during uterine damage and repair we also examined Blimp1 expression following RU486 mediated termination of pregnancy. In RU486 treated wild type females expression is highly up-regulated in peripheral decidual cells prior to their detachment from the uterine wall and release of the decidual mass into the lumen. Expression is subsequently observed at sites of uterine repair and re-epithelialization. By contrast Blimp1 deficient decidual tissues remain fully attached to the uterine wall following RU486 administration. Our transcriptional profiling experiments strongly suggest that Blimp1 suppresses BMP signalling that normally precedes decidual detachment. Additionally Blimp1 is required for up-regulated expression of the extra-cellular metalloproteases that likely mediate RU486 induced tissue shedding.

Interestingly we found in the wild-type post-partum uterus that Blimp1 expression is localized to the stroma immediately adjacent to the over-lying luminal epithelial layer and is strongly up-regulated at the sites of placental detachment undergoing repair and re-epithelialization. Similarly at early post-partum stages we observe robust induction of Mmp10 expression in the stroma at sites of placental detachment as well as along the length of the uterine epithelium. Conditional removal of Blimp1 at PP day 2 is associated with loss of Mmp10 expression. Moreover loss of Blimp1 expression significantly impairs re-epithelialization of the detachment sites and severely impacts involution and remodelling of the uterine epithelium. Our fate mapping studies suggest that Blimp1+ stromal cells fail to contribute to the luminal epithelium during repair and involution, but rather act to promote the re-epithelialization process. Collectively the present findings demonstrate that Blimp1 expression in the endometrial stroma plays essential roles during uterine remodelling and damage repair.

## Results

### Blimp1 dependent resorption of maternal decidual tissues
We previously reported that Blimp1 functional loss from the uterine stroma and luminal epithelium using the progesterone receptor Cre deleter strain (PRCre) disrupts formation of the PDZ[9] and consequently embryonic development arrests at early post-implantation stages (6.5 dpc). Here we characterize Blimp1 requirements during the process of decidual resolution and uterine repair that normally occurs following embryonic failure. Surprisingly we discovered in Blimp1 PRCre females (hereafter referred to as Blimp1 mutants) at 9.5 dpc, some 72 h after embryonic disintegration, that the uterine horns contained normal numbers of decidual swellings (Fig. 1a). Moreover histological analysis revealed that these deciduae retain a normal cellular architecture with the luminal surface coveed by a discrete layer of cytokeratin (CK) positive epithelial cells (Fig. 1b). Additionally CK staining highlighted the persistence of coherent patches of centrally located embryo-derived trophoblast giant cells (TGCs) that closely resemble the post-mitotic primary giant cells derived from the trophectoderm and ectoplacental cone shortly after implantation (Fig. 1b). Thus Blimp1 deficient uteri fail to clear residual decidual tissues at sites of embryonic failure.

Next we examined uterine horns 23 days after copulation plugs were detected, at the equivalent of PP day 5. Normally in wild type

females uterine tissue is rapidly re-modelled and re-epithelialized at the sites of placental shedding to restore the integrity of the uterine lumen (Fig. 1c). This process of tissue repair, in combination with global remodelling of the myometrium and endometrium in response to reduced levels of circulating progesterone, normally prepares the uterus for implantation by the subsequent round of embryos at around 4 days following birth[4]. In contrast in mutant females, decidual masses encapsulated by luminal epithelium are still present within the uterine horns at the equivalent of PP day 5 (Fig. 1c, Supplementary Fig. 1). The presence of these decidual occlusions leads to an accumulation of fluid causing the horns to adopt a swollen cystic morphology. Moreover CK staining highlights the persistence of residual TGCs. By contrast the architecture of the uterus at intra-implantation sites including the myometrial layer, endometrial stroma, luminal and glandular epithelium appears normal. The decidual remnants, encapsulated in a layer of uterine epithelium, persist at PP d14 and PP d28, the latest stage examined (Supplementary Fig. 2). Histological examination of ovarian tissues shows the presence of corpus lutea and maturing follicles (Supplementary Fig. 2). Mutant females continue to cycle normally and thus the inability to resolve decidual remnants cannot be explained by impairment of the hormonal axis.

### Blimp1 expression during RU486-mediated pregnancy termination
Administration of the abortifacient RU486, a potent antagonist of the progesterone receptor, causes shedding of the implantation sites at early stages of pregnancy[11], that is rapidly followed by remodelling and repair of the damaged uterine tissues. To learn more about Blimp1 function during this process next we examined uterine horns recovered following RU486 administration to 5.5 dpc wild type females (Fig. 2a). As expected[9], in control females robust Blimp1 expression is largely confined to the PDZ surrounding the developing embryo, and decidual cells located predominantly on the antimesometrial side, with only scattered Blimp1+ cells present at the periphery of the deciduum. By contrast 14 h post RU486 treatment when the embryo has disintegrated, and the deciduum lacking its normal compact cellular architecture has become infiltrated by maternal blood cells, the pattern of Blimp1 expression dramatically changes. Expression is retained in the PDZ immediately surrounding the embryo, but markedly down-regulated within the central region of the deciduum, while cells at the peripheral margin of the deciduum display robust ectopic expression.

Similarly administration of RU486 at 9.5 dpc resulting in rapid detachment of the decidual sites, is accompanied by distinctive changes in Blimp1 expression (Fig. 2b). Blocking progesterone signalling results in loss of expression from the spiral artery TGCs within the forming placenta, as well as the zone of secondary decidual tissue immediately surrounding the embryo but in contrast leads to robust de novo Blimp1 induction in the outermost peripheral decidual layer.

Twenty-four hours post RU486 administration at 5.5 dpc, just prior to detachment and release of the decidual mass from the uterine wall into the lumen, large numbers of Blimp1+ cells were present at the mesometrial edges (Fig. 3). Expression was also up-regulated in the immediately adjacent stromal cells. Slightly later at 48 h post RU486 treatment once the decidual tissue within the uterine lumen is largely degraded, we observe Blimp1+ cells adjacent to the sites of detachment undergoing repair and re-epithelization. However one day later (72 h post RU486) once the uterus has become fully re-modelled only occasional Blimp1+ stromal and luminal cells are detectable (Fig. 3).

### Blimp1 expression during non-viable embryo resorption
In mice non-viable embryonic implantation sites arising spontaneously, or triggered by genetic defects leading to embryonic failure, are resorbed in a highly local fashion allowing viable littermate embryos to continue to develop normally. To examine Blimp1

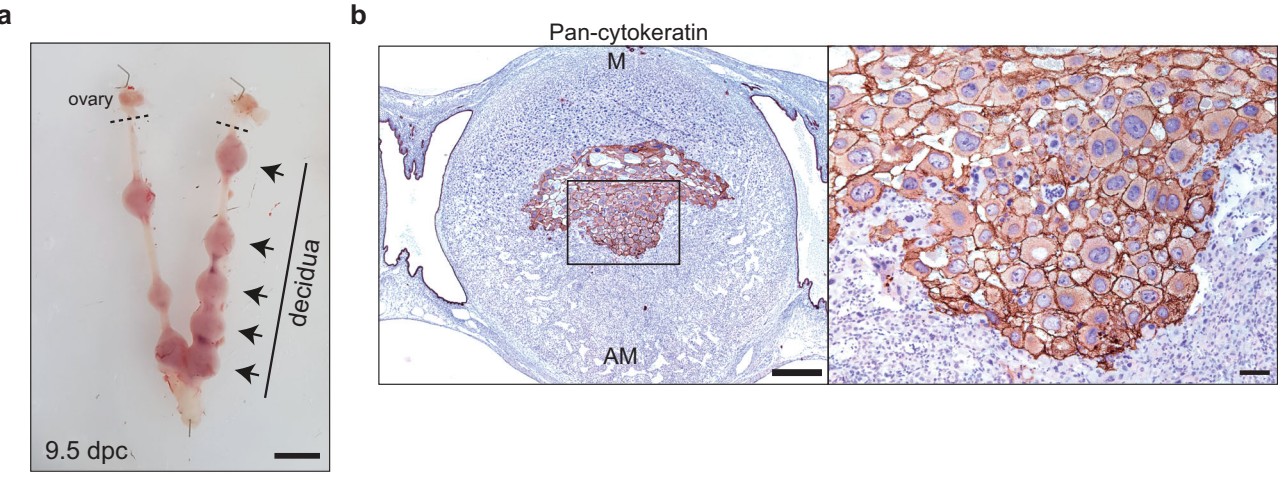

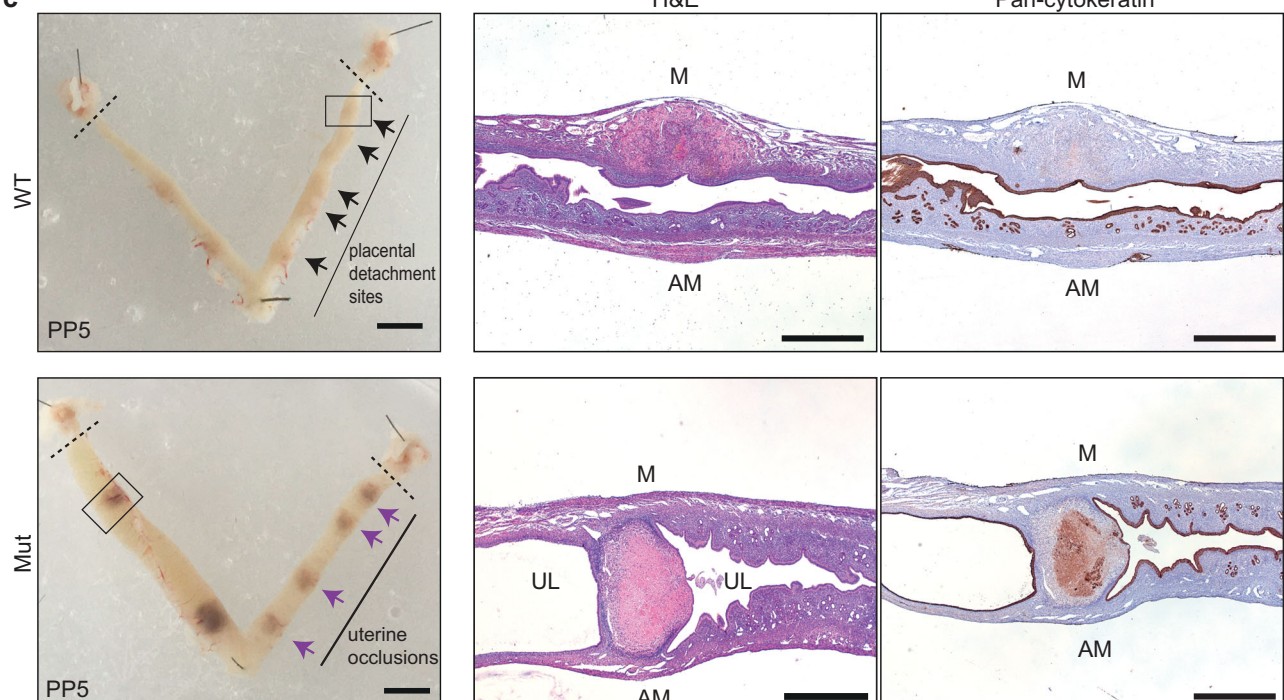

**Fig. 1 | Blimp1 mutants display impaired uterine remodelling at post-partum stages following embryo death. a** Decidual swellings (arrows) are present in 9.5 days post-coitum (dpc) Blimp1 mutant uteri despite the absence of a viable embryo. **b** Epithelialization of the decidual surface and residual parietal trophoblast cells visualised by pan-cytokeratin immunostaining (*n* = 3). **c** The failure of Blimp1 mutants to correctly remodel the luminal epithelium post-partum day 5 (PP5) leads to the formation of cystic structures (purple arrows), flanked by cellular masses containing cytokeratin-positive pTGC-like cells (*n* = 3). Dotted lines distinguish ovaries from the uteri. WT wild type, Mut mutant, UL uterine lumen, M mesometrial, AM antimesometrial. Scale bars = **a**, **c** 4 mm (left panels), 1 mm (right panels), **b** 100 μm, 50 μm (magnified panel).

expression during resorption of defective embryos we made use of a loss of function allele of the T-box gene Eomesodermin (Eomes). Eomes deficient blastocysts implant normally and elicit a decidual response, but arrest shortly thereafter due to Eomes requirements in the trophoblast lineage[12–14]. Deciduae were recovered from heterozygous Eomes intercrosses at 9.5 dpc. At this stage decidual swellings containing homozygous mutant embryos are readily discernable due to their reduced size. Consistent with findings above in RU486 mediated termination of pregnancy, we also observe up-regulated Blimp1 expression at the periphery of resorbing deciduae (Supplementary Fig. 3). These findings demonstrate that up-regulated Blimp1 expression associated with resorption of non-viable embryos occurs independently of a systemic blockade to progesterone signalling.

## Blimp1 mutants are refractory to RU486 induced shedding

Next, to directly test Blimp1 requirements during RU486 induced decidual shedding we injected mutant females at 5.5 dpc. Compared to controls there were no detectable morphological changes 14 h later (Fig. 4a). Similarly 72 h post RU486 injection, when decidual remnants become detached in wild type females (Fig. 4b, c), mutant deciduae remain healthy and fully attached (Fig. 4b, c). A population of CK+ primary TGCs were invariably present (Fig. 4c). Thus Blimp1 mutant females are completely refractory to RU486 mediated elimination of the deciduum and adjacent endometrial tissues.

Previous studies have established a role for enhanced decidual and uterine macrophage and CD4+ T-cell numbers and inflammatory responses following RU486 administration[15]. To examine whether

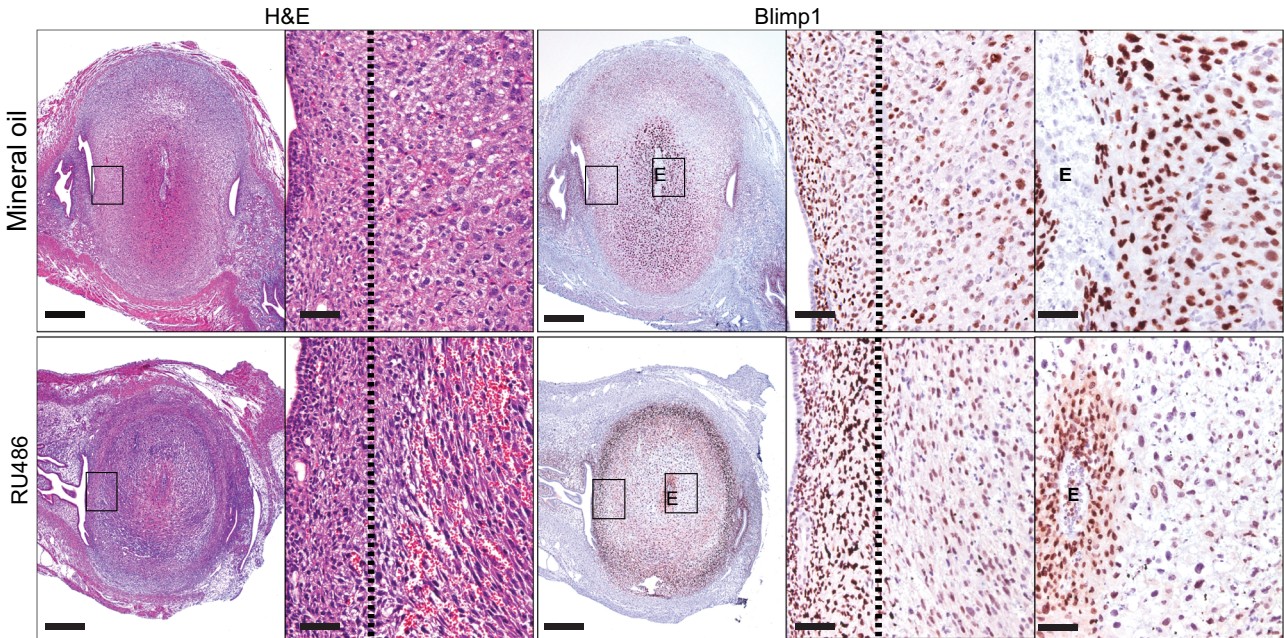

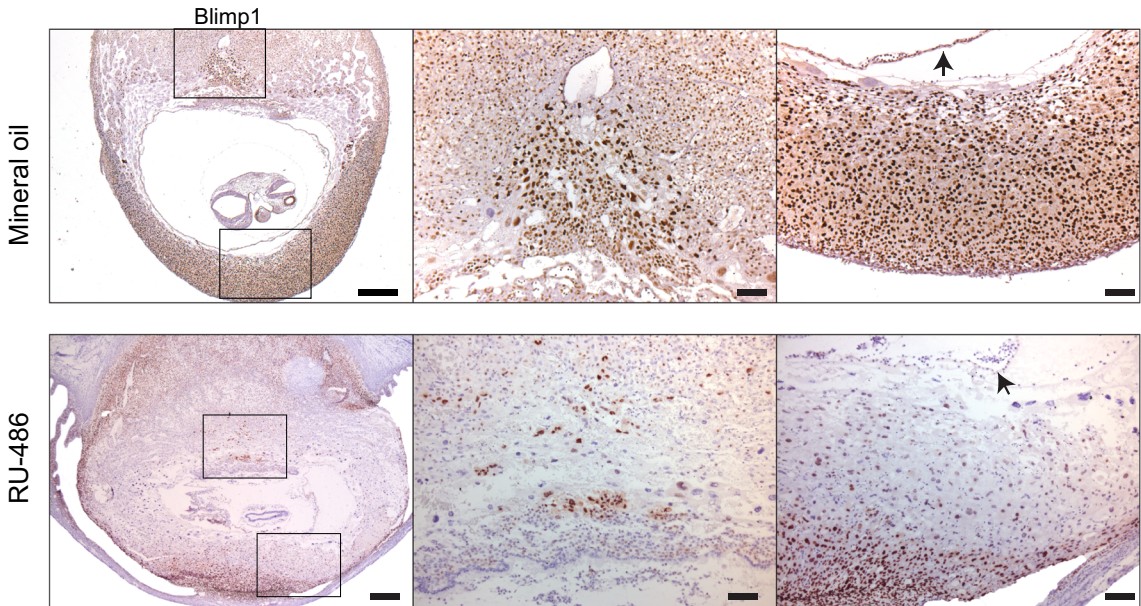

**Fig. 2 | Wild type decidua display dramatic changes in the pattern of Blimp1 expression 14 hours post RU486 treatment. a** Reduced cell-cell contact and accumulation of erythrocytes at the periphery of decidual regions (H&E RU486 panel) in wild type 6.0 dpc decidua ($n = 3$) within 14 h of RU486 treatment (150 μg, IP). Increased Blimp1 expression in the layer of densely packed stromal cells adjacent to these regions. The zone of Blimp1 expression is marked with a dotted line in higher power images. Conversely, Blimp1 immunostaining is reduced in the decidualised stroma surrounding the embryo. **b** By 9.0 dpc RU486 treated wild type decidua ($n = 6$) display more dramatically reduced levels of Blimp1 expression, while displaying markedly increased levels of Blimp1 in the outermost decidual layer. Arrows indicate embryonic visceral yolk sac. E = embryo. Scale bars = **a, b** 100 μm, 10 μm or 50 μm (magnified panels).

reduced immune cell infiltration potentially contributes to the failure of Blimp1 deficient deciduae to detach in response to RU486, we used F4/80 staining to examine the distribution of macrophages. Compared to wild type, similar numbers of macrophages are present in the uterine stroma in the Blimp1 mutant samples. In contrast, as expected[15], we observe numerous infiltrating macrophages in the wild type deciduae undergoing detachment while significantly fewer numbers of macrophages are found within the PRCre deciduae (Supplementary Fig. 4), explaining in part the failure to clear Blimp1 deficient decidual swellings.

### Blimp1 dependent gene expression changes during shedding
To further investigate Blimp1 functional contributions during decidual shedding we performed transcriptional profiling experiments using

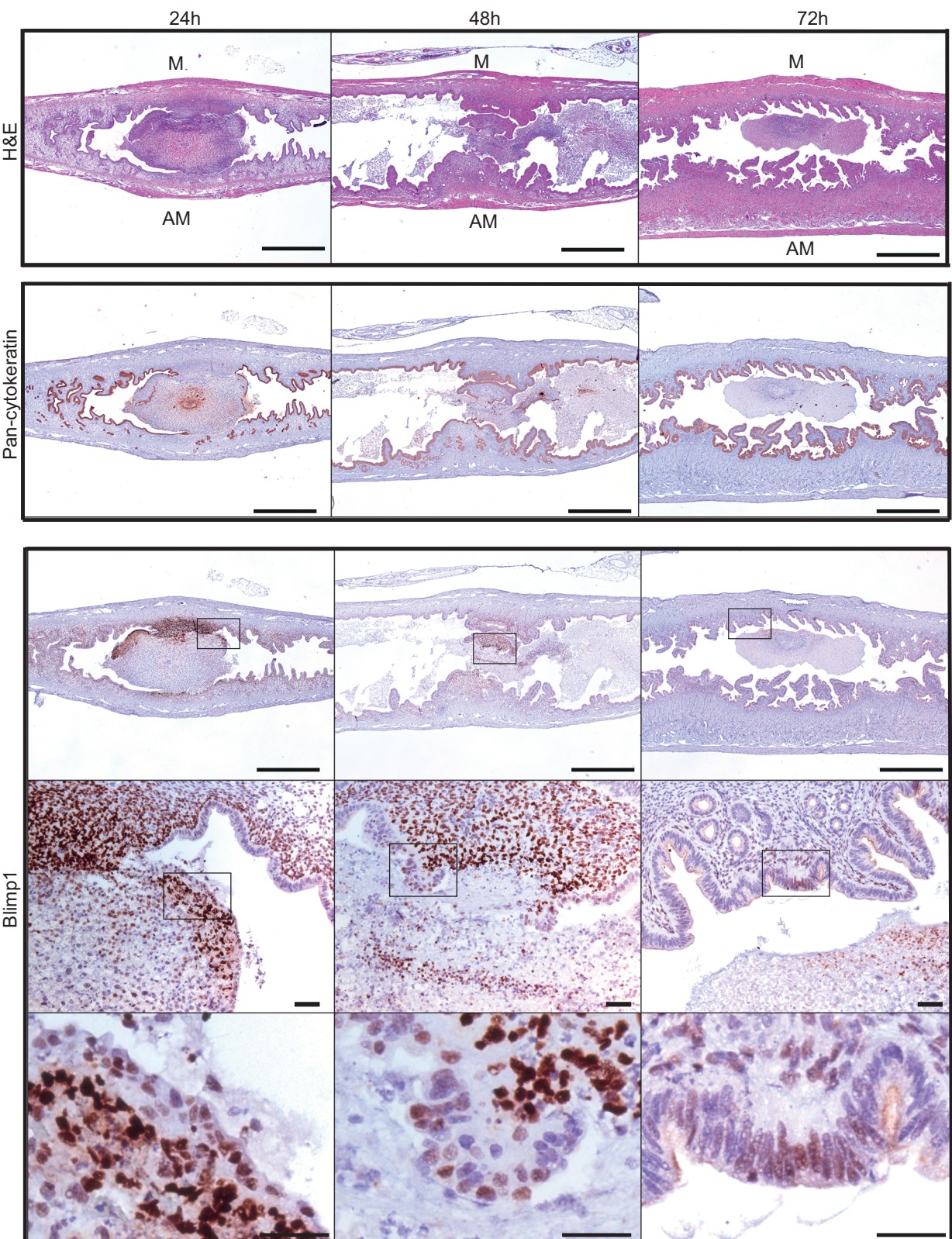

**Fig. 3 | Blimp1 expression in wild type uteri after RU486 administration at 5.5 dpc.** At 24 ($n$ = 3) and 48 h ($n$ = 3) post RU-486, Blimp1 is strongly expressed in stromal and cytokeratin positive luminal epithelial cells at sites of decidual detachment and epithelial repair (black boxes). Higher magnification images of regeneration sites showing nuclear Blimp1 immunoreactivity in the newly forming epithelium. By 72 h ($n$ = 3), Blimp1 is located in few stromal and luminal cells. M mesometrial, AM antimesometrial Scale bars = 1 mm or 50 µm (magnified panels).

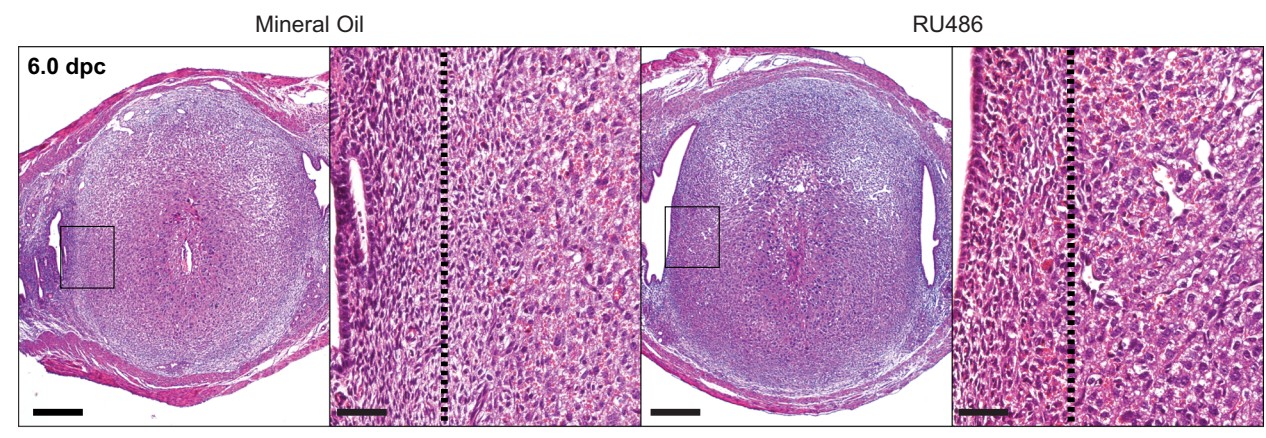

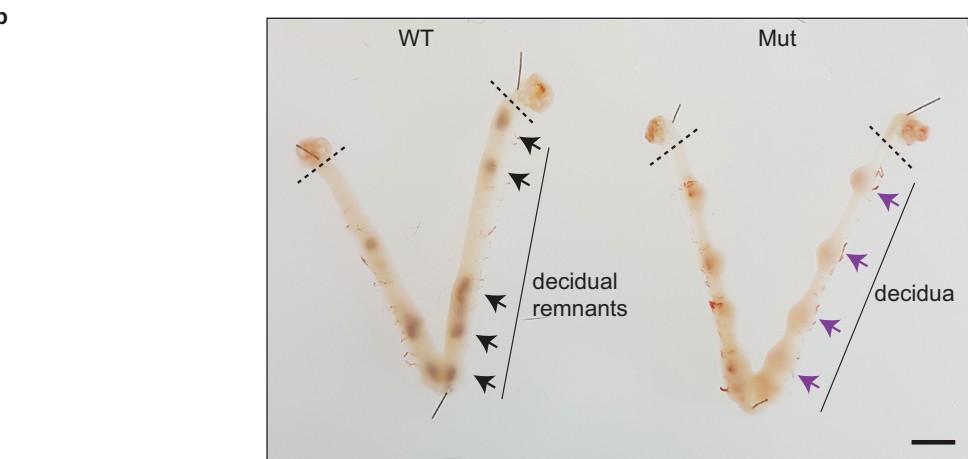

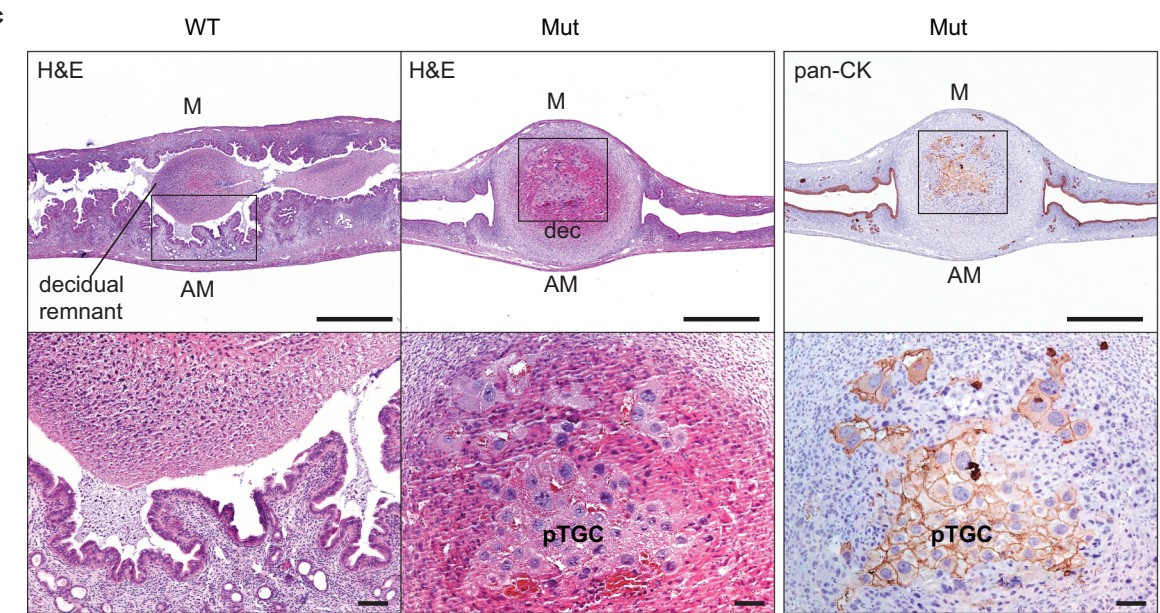

**Fig. 4 | RU486 treatment of Blimp1 mutant females fails to cause decidual detachment. a** RU486 administration fails to induce alterations in cellular architecture. Dotted lines in higher power images mark the equivalent zone of increased Blimp1 expression seen in wild type 6.0 dpc deciduae (*n* = 3) following RU486 treatment. **b** Wild type and mutant uteri 72 h post RU486 administration of 5.5 dpc females. Dotted lines distinguish ovaries from the uteri, while arrows indicate the decidual remnants or deciduae (purple arrows). **c** Histological sections of wild type (*n* = 3) and mutant deciduae (*n* = 3) 72 h post RU486 injection. In Blimp1 mutants, deciduae fail to detach and cytokeratin positive pTGC cells remain detectable. WT wild type, Mut mutant, M mesometrial, AM antimesometrial, dec decidua. Scale bars = **a** 500 μm, 50 μm (magnified panels), **b** 4 mm or **c** 1 mm, 50 μm (magnified panels).

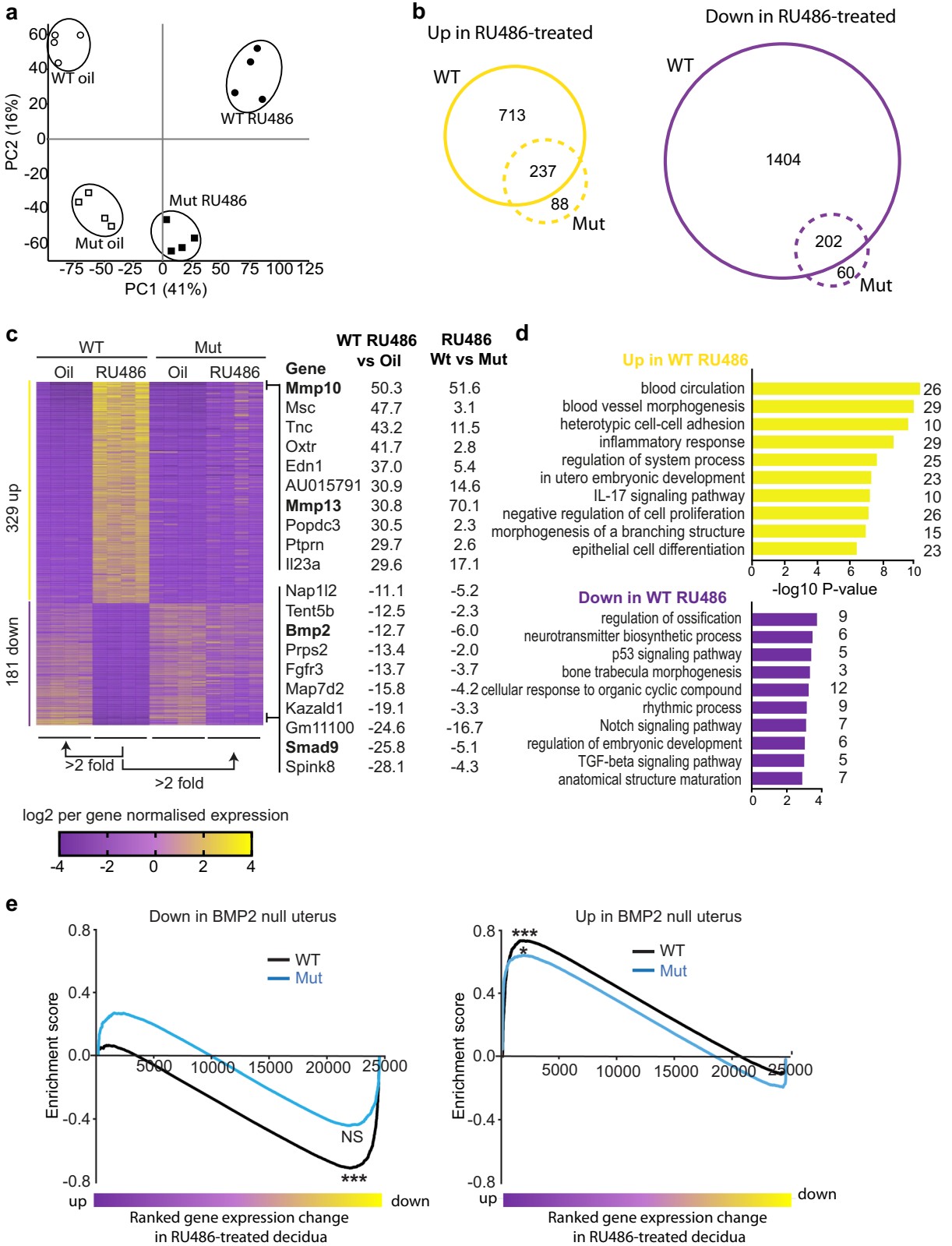

**e**

Down in BMP2 null uterus

Up in BMP2 null uterus

log2 per gene normalised expression

RNA-seq. We compared RU486-treated wild type and Blimp1 mutant decidual tissues. Samples were collected 14 h post-RU486 administration when altered Blimp1 staining patterns and gross morphological abnormalities first become apparent in wild types (Fig. 2a).

As expected[9], Blimp1 conditional inactivation in PR expressing cells alters decidual gene expression, with clear separation of wild type and mutant samples based on principal component analysis,

irrespective of RU486 administration treatment (Fig. 5a). RU486-treatment of wild type females was associated with dramatic changes in gene expression (Fig. 5a). At the level of >2 fold, 950 genes were up- and 1606 genes down-regulated relative to oil-treated controls (Fig. 5b, Supplementary Data 1, 2). Consistent with the absence of overt histological changes, substantially fewer genes were altered in mutant decidua following RU486-treatment (Fig. 5a, b, Supplementary Data 1,

**Fig. 5 | RNA-Seq analysis of mutant and wild type decidua 14 h post-RU486 treatment. a** Principal component analysis of wild type and mutant decidual samples 14 h post treatment with oil vehicle or RU486. RU486 induced transcriptional changes are greater in wild type samples compared to mutants relative to their respective oil-treated controls. Blimp1 genotype also impacts on gene transcription as previously reported[9]. **b** Venn diagram overlaps of altered genes (>2 fold) following RU486 treatment in wild type and Blimp1 mutant decidual samples. Fewer genes are altered in Blimp1 mutant decidua following RU486 treatment. Of those that do change, the majority overlap with those that also change in RU486 treated wild type samples. **c** Heatmap of genes up (n = 329) and down-regulated (n = 181) > 2 fold in wild type decidua following RU486 treatment, that are also >2 fold different compared to mutant RU486-treated samples. The top 10 up and down regulated genes are indicated, with genes of special interest highlighted in bold. **d** Gene ontology analysis (www.metascape.org) of the 329 up and 181 down-regulated genes from panel c. *P*-values were generated by the program using a hypergeometric test without multiple testing correction. **e** GSEA comparing DESeq2 directional fold and FDR-ranked genes in E6.0 decidual Blimp1 mutant and wild type decidua, 14 h post-RU486 treatment with Bmp2 target genes[20]. Down-regulated gene expression in Blimp1 wild type but not Blimp1 mutant decidua following RU486 treatment significantly correlates with genes downregulated in Bmp2 null uterus. In contrast, upregulated gene expression in Blimp1 wild type and Blimp1 mutant decidua following RU486 treatment both significantly correlates with genes upregulated in Bmp2 null uterus. NS non-significant, *FDR *q* = 0.011. ***FDR *q* = 0.000.

2). At the level of >2 fold, 325 genes were up-regulated and 262 genes down-regulated compared to oil-treated controls.

We identified 510 RU486-responsive genes in wild type decidua that show >2 fold difference as compared to RU486-treated mutant samples (Fig. 5c). GO term analysis reveals that genes associated with multiple processes and pathways are selectively altered in RU486-treated wild type decidua (Fig. 5d). Consistent with previously identified roles in decidual breakdown and/or uterine repair[16,17] up-regulated genes include those involved in blood vessel morphogenesis and epithelial cell differentiation.

Among the top 10 selectively down-regulated genes in wild type decidua following RU486 treatment are two members of the TGF-β signalling pathway, namely Bmp2 and Smad9, an intracellular mediator of BMP signalling[18]. BMP signalling plays an important role during stromal cell decidualisation in both mice and humans[19]. Moreover, Bmp2 signalling has been shown to be essential for the maternal decidual response in mice[20]. Previous studies demonstrate that Bmp2 expression is dependent on PR signalling since Bmp2 protein levels are depleted in decidual samples following RU486-treatment[21].

Genes down-regulated in WT decidua following RU486 treatment also include known targets of TGF-β signalling such as Fst, Id1 and Id3[18]. Interestingly, comparison of all 501 genes selectively altered in wild type decidua following RU486 treatment with 5529 curated gene sets within the MsigDB gene database revealed significant overlap with genes altered in Bmp2 null deciduoma (BMP2_TARGETS)[20,22]. Notably, genes selectively down-regulated in WT decidua following RU486-treatment overlap with those down-regulated in Bmp2 null deciduoma and vice versa (Fig. 5e). Thus, a subset of mis-regulated genes in RU486-treated wild type decidua are likely altered as a secondary consequence of loss of Bmp2 signalling.

To directly examine the impact of RU486 treatment on BMP signalling next we performed immuno-staining experiments. As expected 6.5 dpc wild type decidual samples display widespread nuclear phospho-Smad1/5/9 staining, predominantly in the secondary decidual zone (Fig. 6a). Within 14 h of RU486 treatment, co-incident with down-regulation of Bmp2, phospho-Smad1/5/9 staining is reduced to barely detectable levels in wild type decidua. In Blimp1 mutants strong nuclear staining is detectable throughout the decidual tissue including cells immediately surrounding the embryo. However, in marked contrast to wild type, there is no apparent change in the intensity or distribution of pSmad1/5/9 staining following RU486 treatment. We speculate that this failure to down-regulate BMP/Smad signalling may contribute to the survival of empty deciduae that persist in mutant females.

Among the top 10 genes preferentially up-regulated in RU486-treated wild type decidua are the two extracellular matrix degrading enzymes, Mmp10 and Mmp13. Mmp family members are known to play key roles in tissue remodelling[23]. Mmp10, also known as Stromelysin II, has broad substrate specificity with potential to degrade multiple components of the extracellular matrix[24]. Previous studies have shown Mmp13 is selectively induced at sites of tissue breakdown, initially at the interface of the basal and decidual zones in a mouse model of endometrial breakdown and repair in response to loss of progesterone signalling[25].

To further describe changes in Mmp10 expression resulting from RU496 administration we performed immuno-staining experiments (Fig. 6b). There was no detectable Mmp10 expression in control oil treated samples within the decidual stroma. However 14 h post-RU486 injection, the peripheral tissue layer as well as the decidual cells immediately surrounding the implantation site contain abundant levels of secreted Mmp10. Strikingly these domains of Mmp10 expression closely mirror those that ectopically up-regulate Blimp1 expression (Fig. 2a). In contrast, and consistent with the transcriptional profiling data, RU486 treatment of Blimp1 mutant females fails to cause up-regulated Mmp10 expression in the peripheral decidual tissues. This observation likely explains in part why mutant deciduae fail to undergo degradation and detach from the uterine wall.

## Blimp1 and Mmp10 co-expression during uterine remodelling

In wild type females following expulsion of the placenta at birth, uterine lesions at sites of placental detachment on the mesometrial side of the uterus are rapidly resolved and the integrity of the epithelium is normally restored within 3 days. This process of tissue repair is associated with alterations in cell death and proliferation of the luminal and stromal cell compartments immediately following parturition[26]. To learn more about Blimp1 functional contributions next we investigated the pattern of Blimp1 expression during this remodelling process. At PP day 2 Blimp1+ cells are present in the stroma directly adjacent to the collapsed and folded epithelium along the length of the uterine horns (Fig. 7a). Moreover cells that robustly express Blimp1 are also present at the sites of placental detachment both within in the stroma as well as the population of cells on the denuded luminal surface undergoing repair and re-epithelialization by CK+ cells. Since decidual shedding following RU486 administration is associated with up-regulation of Mmp10 (Fig. 6b), next we examined the pattern of Mmp10 at PP day 2. Strikingly abundant levels of secreted Mmp10 co-localize with the Blimp1+ stromal cell population both at the placental detachment sites as well as the cells immediately adjacent to the epithelium along the length of the uterine horns (Fig. 7b).

By PP day 5, when the uterus has almost fully completed involution and remodelling to return to the pre-pregnancy stage in preparation for embryo implantation, Blimp1+ cells are confined to the stroma underlying the epithelium. By contrast Mmp10 expression levels in the endometrium have significantly declined with only residual levels of immunoreactivity confined to the placental detachment sites (Supplementary Fig. 6).

## Blimp1 is required for post-partum uterine involution

To examine whether Blimp1 has essential roles in the process of tissue remodelling and restoration of luminal epithelial integrity, we used a genetic approach to eliminate Blimp1 activity from the immediate post-partum uterus. We generated females carrying a loxP-flanked Blimp1 conditional allele[27] in combination with the Prdm1/Blimp1-

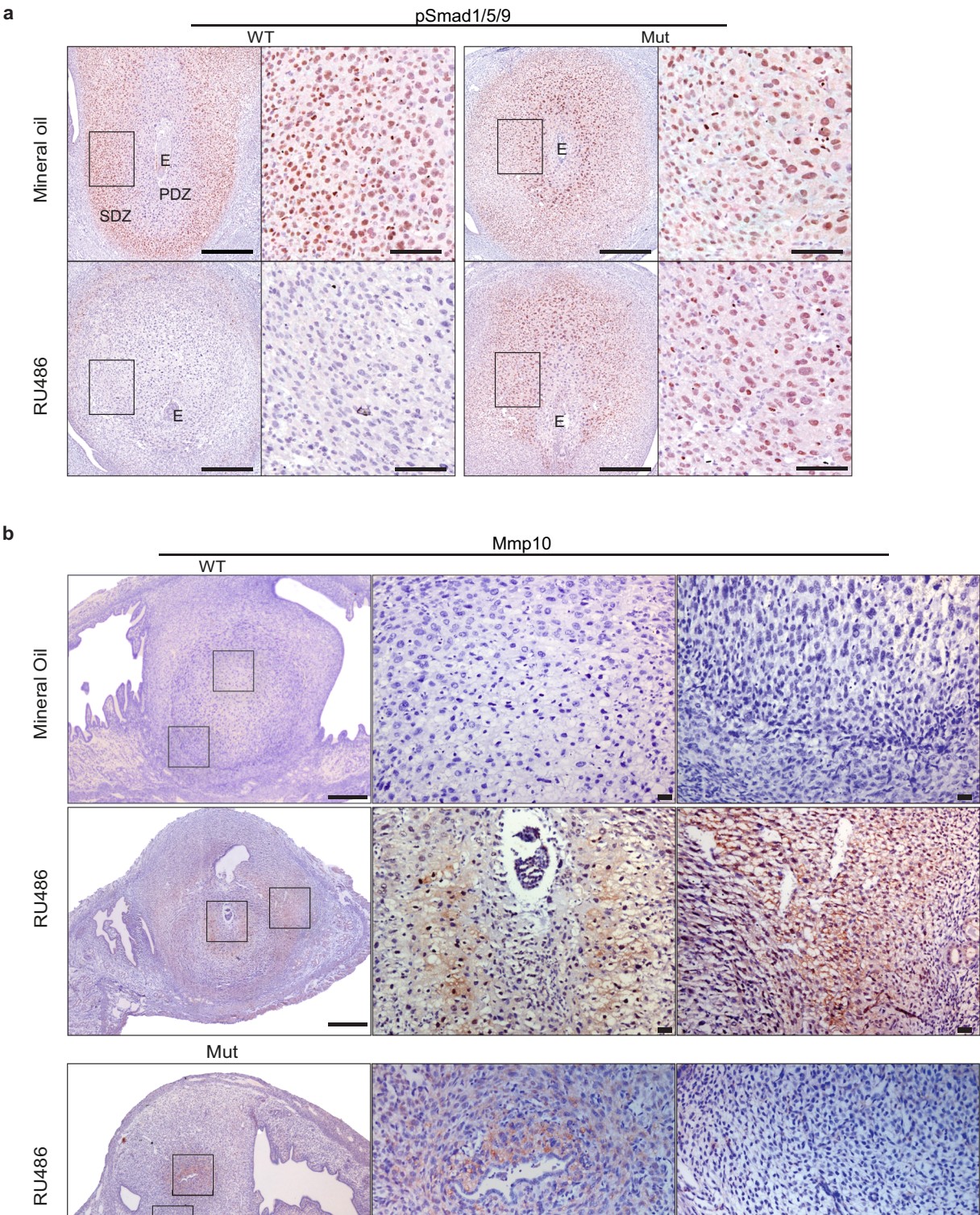

**Fig. 6 | PSmad1/5/9 and Mmp10 expression in 6.0 dpc decidual samples 14 h post RU486 treatment. a** In wild type controls (*n* = 3), nuclear Smad1/5/9 staining is predominantly observed in the SDZ, whereas in mutants (*n* = 3), the cells immediately surrounding the embryo are also positive. Staining is substantially reduced in RU486-treated WT but not mutant. **b** Compared to wild type controls (*n* = 4), Mmp10 is strongly up-regulated in RU486 treated wild type decidual cell populations immediately adjacent to the embryo implantation site and in the peripheral layer. Expression is unaffected in RU486 treated mutant samples (*n* = 3). PDZ primary decidual zone, SDZ secondary decidual zone, E embryo. Scale bars = **a** 500 μm, 50 μm (magnified panels), **b** 500 μm, 100 μm (magnified panels).

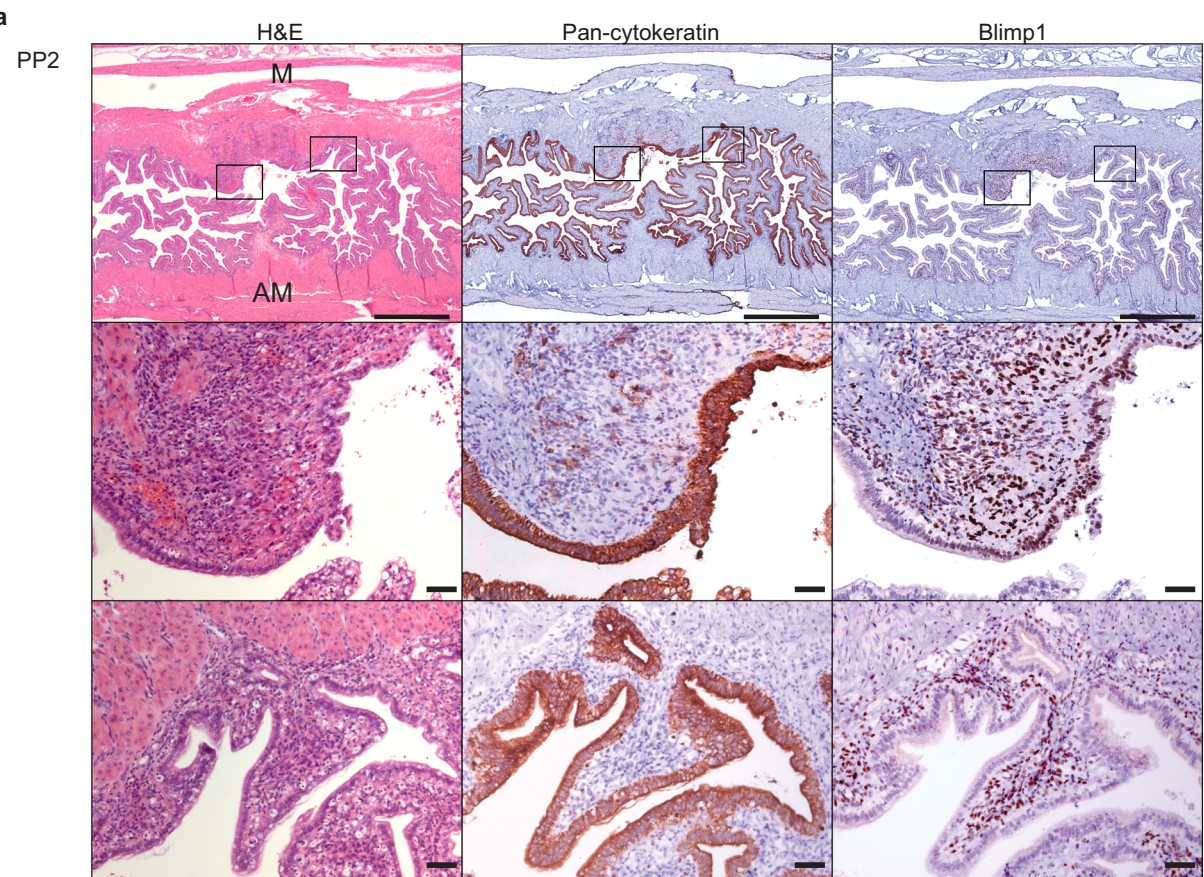

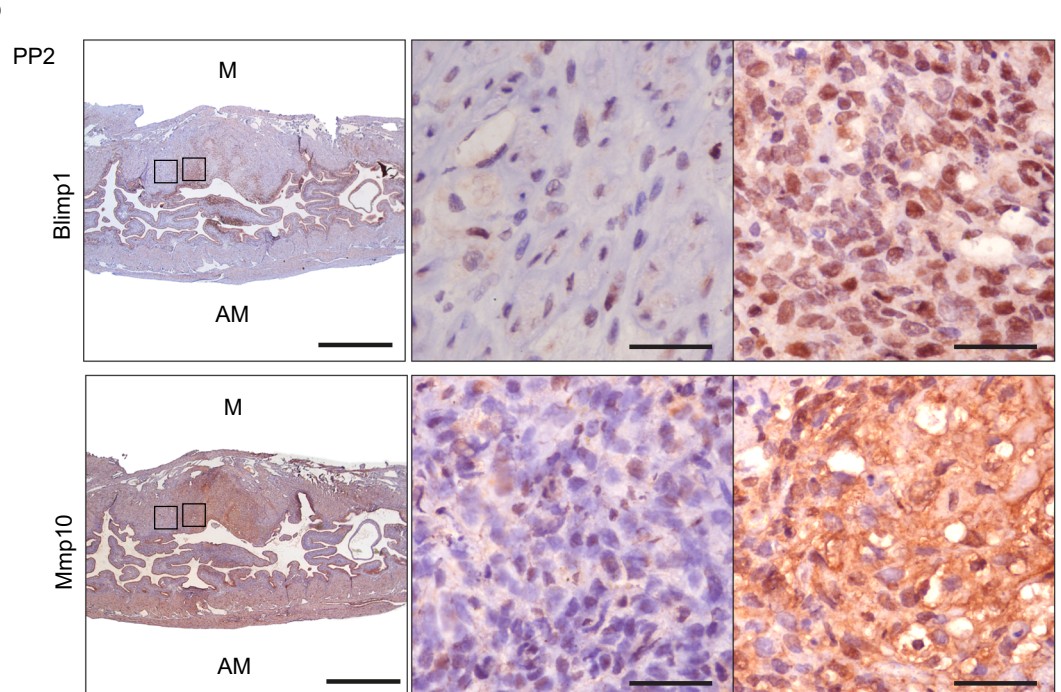

**Fig. 7 | Blimp1 and Mmp10 are co-expressed in wild type uteri during early post-partum regeneration. a** Blimp1 is expressed in stroma cells and regenerating luminal epithelium at PP2 at sites of placental detachment (*n* = 7). **b** Overlapping areas of Blimp1 and Mmp10 expression in PP2 uteri. Areas of low (mid panels) and high (right panels) Blimp1 and Mmp10 expression are displayed (*n* = 7). M mesometrial, AM antimesometrial. Scale bars = 1 mm, 50 μm (magnified panels).

Cre^(ERT2) allele[28]. We determined that a single injection of tamoxifen (Tx) within 6 hours of parturition results in very efficient loss of Blimp1 immunoreactivity in the stromal cell population at PP day 2 (Supplementary Fig. 5). Consequently, at PP day 5 and 7 Blimp1 expression is largely undetectable in the stromal cell populations immediately adjacent to the luminal epithelium (Supplementary Fig. 5).

As judged by histological criteria, the Blimp1 deleted (Blimp1Δ) uterine horns fail to efficiently repair the placental detachment sites at

PP day 2. We observe considerable residual tissue damage, associated with the accumulation of blood and acellular debris within the lumen and significant localized tissue sloughing (Fig. 8a). This phenotype is associated with defective epithelial repair as highlighted by cytokeratin staining. Moreover loss of Blimp1 activity results in a failure to upregulate Mmp10 expression (Fig. 8a).

In wild type females epithelial repair and involution restores the uterus morphology to the pre-pregnancy state by PP day 5. In contrast at this stage, while the surface of the detachment sites has become re-epithelialized, and Mmp10 levels resemble that of wild type, the Blimp1Δ uterine horns have not remodelled appropriately and the surface area of the uterine lumen and thickness of the underlying endometrium fails to be significantly reduced (Supplementary Figs. 5 and 6). This impairment to involution in the absence of Blimp1 expression is even more striking at PP day 7 stages (Fig. 8b). Defective involution is not caused by Tx since the morphology of uterine horns from wild type females similarly injected with Tx is indistinguishable from controls (Supplementary Fig. 7). These results demonstrate that Blimp1 expression at parturition is essential to promote both epithelial repair and timely involution of the uterus.

### Fate mapping Blimp1 stromal cells during tissue repair

Previous studies have pointed to the existence of sub-sets of long lived uterine stem cells that contribute to the regeneration of the epithelium, endometrium and myometrium during post-partum regeneration[4,5]. Gene-based lineage tracing experiments have identified putative epithelial progenitors within the stroma that contribute to the luminal epithelium during involution[26,29]. Here we observed Blimp1+ cells are closely intermingled with the cytokeratin positive epithelium at sites of re-epithelialization (Fig. 7). To test whether these cells potentially contribute to the re-forming epithelium we performed fate mapping experiments. Females heterozygous for the Prdm1CreERT2 allele and also carrying either the Rosa26R^YFP or Rosa26R^LacZ Cre recombinase-activated constitutive reporter alleles[30,31] were injected with Tx at PP day 0 or day 2 respectively. Tx-mediated nuclear accumulation of Cre recombinase results in the excision of the reporter allele and indelibly marks the descendants of the Blimp1+ stromal cell population. The distribution of labelled GFP or LacZ+ cells was examined at PP day 5 when epithelial repair and remodelling has largely completed. We found that the descendants of Blimp1+ cells remain confined to the stromal population along the length of the uterus (Fig. 9). Thus we conclude that essential Blimp1 activities during tissue repair and remodelling of the luminal epithelium are restricted to the stromal cell compartment, and that this cell sub-set does not directly contribute to regeneration of the epithelial layer.

## Discussion

The uterus displays a remarkable capacity for tissue remodelling during the oestrus cycle, implantation and pregnancy, and subsequently during post-partum involution. In mice substantial cyclical remodelling of the epithelium and endometrium[2] is driven by the activity of distinctive epithelial and stromal stem cell populations identified via genetic lineage tracing experiments[4,5,32,33]. In other mammals, including humans, cycling is accompanied by shedding of the uterine endometrium via the process of menstruation. During menstruation and following parturition the damaged endometrium is rapidly regenerated to restore uterine integrity and tissue architecture[4,5].

Here we demonstrate that the zinc finger transcription factor Blimp1 plays an essential role in decidual tissue during pregnancy. Blimp1 inactivation within the highly specialized decidual tissue via PR-Cre mediated deletion renders it refractory to the process of breakdown that normally follows embryonic failure. Rather the decidual swellings, devoid of the embryo proper, and containing only pockets of residual post-mitotic primary TGCs, persist fully 3-4 weeks after

their original induction. Thus Blimp1 appears to have essential roles in mediating localized resorption and uterine remodelling normally triggered in response to embryonic loss during gestation. By contrast Blimp1 is not required for hormonally regulated endometrial remodelling and homeostasis during the oestrus cycle since virgin Blimp1 PR-Cre mutant females cycle normally and ovulate appropriately[9]. However following the first pregnancy decidual tissues fail to be shed and resorbed, and rather become encapsulated by luminal epithelium leading to occlusion of the uterus. The presence of these long-lived decidual masses has no impact on subsequent oestrus cycles, but prevents subsequent fertilization of released ova.

Remarkably we discovered that Blimp1 deficient decidual tissues fail to detach from the uterine wall and are completely resistant to RU486 treatment. It is well known that blocking progesterone activity results in abrupt termination of pregnancy associated with the rapid loss of embryo viability and sloughing of the associated maternal tissues. In the mouse this process is initiated within a few hours following RU486 administration. We exploited this model to investigate Blimp1 functional contributions during gestational uterine remodelling and repair. Our experiments demonstrate a dramatic alteration in the distribution of Blimp1+ decidual cells in response to RU486 treatment. With 14 h of administration at either at 5.5 dpc or 8.5 dpc, cell populations located at the periphery of the deciduum show robust up-regulated Blimp1 expression. Blimp1 expression persists in the mesometrial decidual cells 10 h later following release of the residual decidual mass from the uterine wall. Blimp1 expression is also locally retained in the stroma and epithelial cell layer undergoing repair.

Previous studies have established that Bmp2 plays a critical role during decidualization in mice[20] and that RU486 treatment specifically results in loss of Bmp2 protein[21]. Interestingly we found here that retention of the Blimp1 deficient decidual swellings is associated with persistent BMP signalling. In wild type females RU486 treatment causes a dramatic down-regulation of Bmp2 and Smad9 transcription and loss of pSmad 1/5/9 staining throughout the decidua. Our transcriptional profiling experiments show that genes down-regulated in RU486 treated wild type deciduae overlap with those known to be down-regulated in Bmp2 null deciduae. By contrast BMP signalling is retained throughout the decidual tissue in RU486 treated Blimp1 mutants suggesting that enhanced BMP signalling contributes to the survival of the mutant decidual cell population. Recent single-cell profiling studies have shown that primary TGCs express multiple TGFβ-signalling pathway components including BMP receptors[34]. Thus increased BMP signalling observed here in the absence of Blimp1 probably contributes to the persistence of TGC within the decidual swellings.

In RU486 treated wild type females expression of two metalloproteases namely Mmp10 and Mmp13, that actively degrade components of the extra-cellular matrix is rapidly up-regulated. Mmp13 has previously been implicated in a model of endometrial breakdown and repair[25], while Mmp10 mRNA levels rise dramatically in the stroma and epithelium at birth[35]. Mmp13 and Mmp10 up-regulation at the end of pregnancy correlates with a marked decline in progesterone levels in the maternal circulation. Moreover siRNA mediated silencing of the progesterone receptor in human myometrial cells results in dramatically increased Mmp10 mRNA expression[36].

In wild type females robust levels of Mmp10 protein are detected in peripheral decidual cells and in the PDZ within 14 h of RU486 administration. Notably these sites correspond to those in which Blimp1 is up-regulated. Blimp1 deficient deciduae fail to up-regulate Mmp10 in response to blocking progesterone receptor signalling. These results strongly suggest that induction of Blimp1 expression in response to progesterone receptor blockade is required either directly or indirectly for up-regulation of these important remodelling proteases to promote decidual shedding in the wild type uterus.

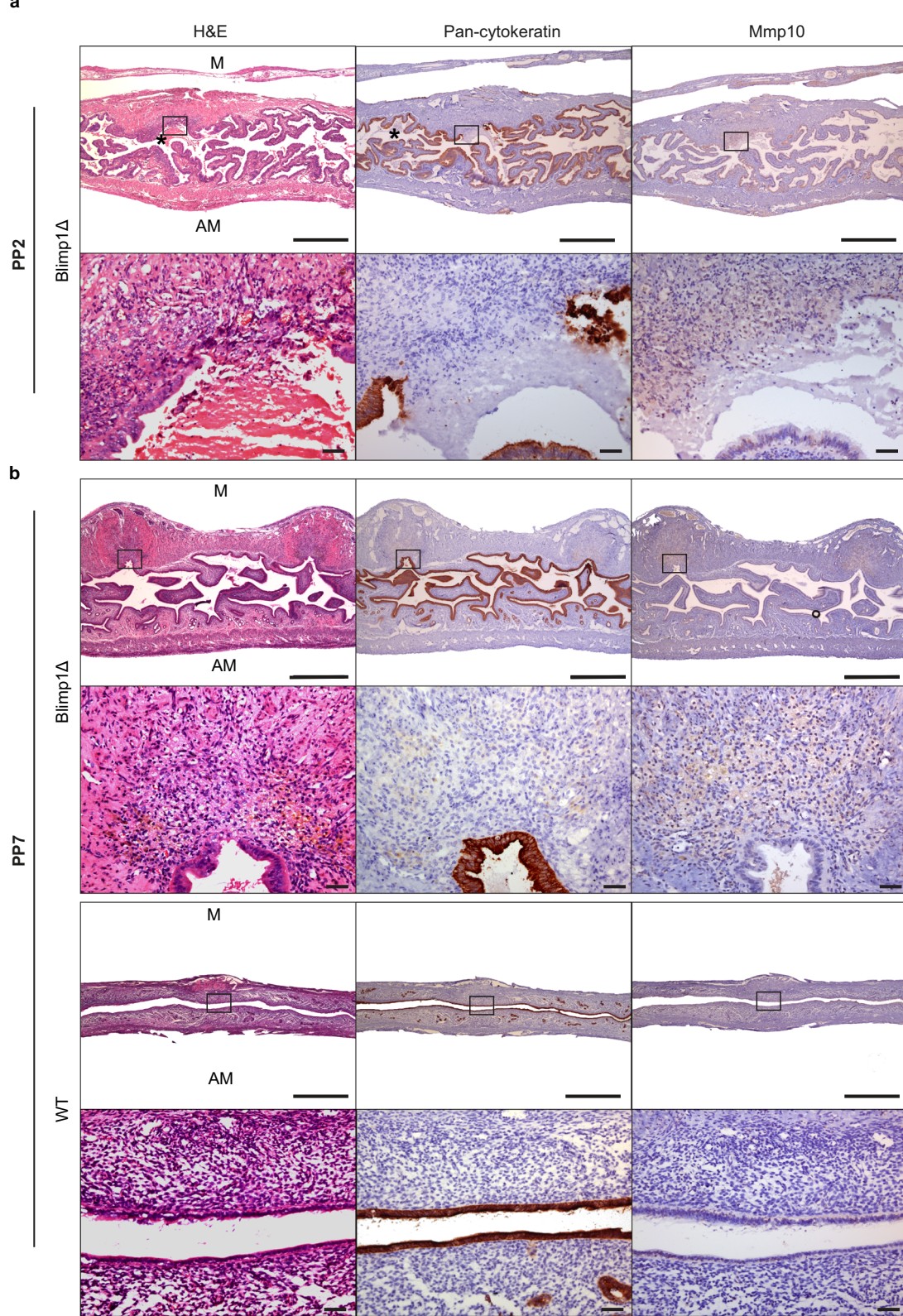

**Fig. 8 | Abnormal uterine repair in Blimp1Δ females. a** At PP2 in Blimp1Δ uteri, debris and blood (asterisks) are observed. Mmp10 staining is dramatically reduced whereas re-epithelization of the detachment site has not yet been completed (*n* = 5). **b** By PP7, wild type uteri have been fully repaired and remodelled whereas the endometrium in the mutants fails to appropriately remodel and residual Mmp10 staining is confined to the placental detachment sites (*n* = 3). M mesometrial, AM antimesometrial. Scale bars = 1 mm, 50 μm (magnified panels).

a

PP0 or PP2 i.p. injection Tx

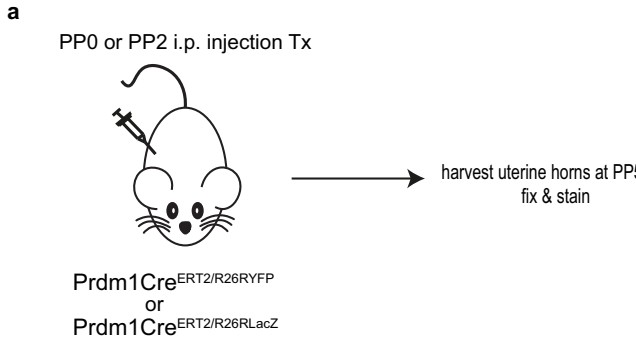

harvest uterine horns at PP5,
fix & stain

Prdm1Cre$^{ERT2/R26RYFP}$
or
Prdm1Cre$^{ERT2/R26RLacZ}$

b

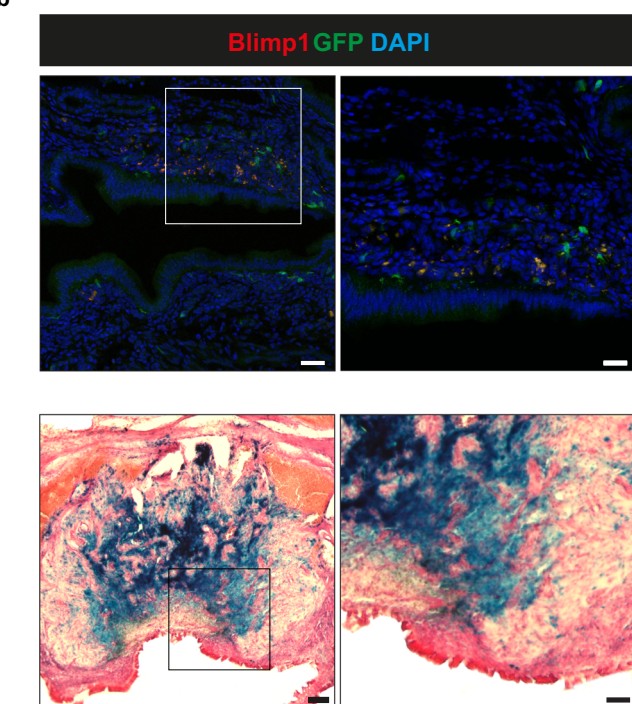

**Blimp1** GFP DAPI

**Fig. 9 | The Blimp1 expressing stromal cell population does not contribute to newly formed luminal epithelium. a** *Prdm1Cre* $^{ERT2/+}$;*R26R* $^{YFP/+}$ (*n* = 3) or *Prdm1Cre* $^{ERT2/+}$;*R26R* $^{lacZ/+}$ (*n* = 4) females were injected i.p. with Tx at PP0 and PP2 respectively, uteri harvested at PP5 and processed for staining. **b** Cryosections of detachment sites stained for Blimp1 (red), GFP (green) or LacZ (lower panel) are displayed, focusing on the luminal epithelium and the adjacent stroma. Scale bars = 500 μm (upper panel), 1 mm (lower panel), 50 μm (magnified panels).

Interestingly, as for sites of RU486 induced decidual shedding during gestation, we find robust induction of Blimp1 expression within the uterine stroma following parturition. Blimp1 is broadly expressed at the sites of tissue damage associated with release of the individual placental structures. Moreover Blimp1+ expression demarcates a subset of the stromal population lying immediately adjacent to the luminal epithelium along the length of the uterine horns which persist until at least PP day 7. Here we find that conditional removal of Blimp1 activity from the post-partum uterus significantly impacts the process of repair and involution. By day 2, in the areas of the lumen denuded of epithelium as a result of placental expulsion, the process of tissue repair is significantly impaired, leading to a delay in re-epithelization and inappropriate release of blood and cellular debris into the uterine lumen. Generalized loss of Blimp1 activity from the stroma also leads to a failure to reduce and remodel the expanded epithelium and endometrium which normally occurs within the first 5 days after parturition. Blimp1 is thus essential to both promote the re-

epithelialization process and restore the endometrium in preparation for embryo implantation.

The process of re-epithelization of the mouse uterus has been investigated extensively and a number of differing models proposed[4,5]. Genetic fate mapping studies support a mechanism whereby the epithelial population is repaired by self-renewal[4,5], while other experiments suggest epithelial regeneration occurs in part via a process of mesenchyme-to-epithelial transition by a sub-set of stromal cells[26,29,37,38]. Consistent with this latter possibility at PP day 2 we find islands of Blimp1+ stroma adjacent to denuded areas resulting from placental detachment and which are undergoing rapid re-epithelialization. However our fate mapping experiments conclusively establish that by PP day 5 the Blimp1+ cells do not give rise to the newly repaired epithelial surface but rather are confined to the stroma immediately underlying the epithelial surface, and thus do not represent a multipotent progenitor cell population.

Blimp1 deficient uterine horns are blocked in their ability to undergo normal involution, the process that returns the entire organ to the pre-pregnancy state. Very few studies have assessed the cellular mechanisms underlying involution, with most focused on identifying potential stem cell populations within the myometrium and endometrium responsible for cyclical repair and remodelling[4,5]. At the end of pregnancy, the endometrium and epithelial surface has expanded several fold but after birth resolves within a 5-day period. This coordinate and highly dynamic diminishment in tissue mass occurs via a poorly understood process. Here we have uncovered a key transcription factor Blimp1 expressed within the stroma underlying the uterine epithelial layer that orchestrates resolution of the endometrium and associated luminal epithelium. Degradation effectors including multiple matrix metalloproteases are known to be induced in response to the fall in progesterone levels that occur at parturition[35,39,40]. Additionally in the human uterus multiple metalloproteases are induced during post-menstrual endometrial regeneration[41]. Our current studies have established that both upregulated expression of Mmp10 and Mmp13 fails to occur in the absence of Blimp1. However to date single loss of function mutations in for example Mmp7 and Mmp3[39], also known to be upregulated in the post-partum uterus, as well as either loss of Mmp10 or Mmp13 have no impact on uterine involution or female fertility[42,43]. An attractive possibility that warrants future investigation is that Blimp1 activity is required to coordinate the expression of a network of tissue remodelling genes necessary to drive both repair and involution during uterine homeostasis. Given the apparent conservation of uterine repair mechanisms between humans and mice[4,5] it will be interesting to examine possible Blimp1 involvements in human menstruation and post-partum uterine remodelling.

Since its original identification as a silencer of β-interferon gene expression[44] and a master regulator of plasma cell maturation[45], Blimp1 has been shown to govern development within numerous additional tissues, including specification of the mammalian germ line, maturation of the gut epithelium, mammary gland homeostasis, and formation of the placenta during pregnancy[46,47]. Its activity as a transcriptional repressor during B-cell maturation involves recruitment of chromatin-modifying factors such as histone deacetylases and methyl transferases[46]. However the possibly common mechanisms by which it regulates target gene expression in these distinct tissue contexts remain largely unknown. Here we have discovered an essential role for Blimp1 during uterine repair and involution. An important future goal is to identify Blimp1 transcriptional targets and the associated gene regulatory networks that underpin the critical roles it plays in the context of uterine remodelling.

## Methods

### Animal care and use
All animal experiments were performed in accordance with the UK Home Office regulations and approved by the University of Oxford

Local Ethical Committee. Animals were maintained under a 12 h light/dark cycle and housed in individual ventilated environmentally controlled cages (20 °C, 55% humidity). C57BL/6 female mice were used as wild type controls (6–10 weeks of age). To conditionally inactivate Blimp1 in the uterus Prdm1BEH/+:PRCre/+ males[6,9,10] were crossed to Prdm1CA/CA mice[27] to generate Prdm1BEH/CA;PRCre females (referred to as Blimp1 mutants throughout the text). Genotyping was performed as described in the original reports. Both wild type and mutant females were mated to wild type C57BL/6 males, and the day of finding the vaginal plug was designated as day 0.5 dpc. Eomes[+/−] mice[48] were intercrossed to generate deciduae carrying non-viable Eomes[-/-] embryos. For the RU486 experiments pregnant females were injected intra-peritoneally (i.p.) with 150ug of RU486 (ab120356, Abcam), or mineral oil (M5904, Sigma-Aldrich) as a control, and sacrificed at defined post-injection time points. To assess Blimp1 contributions in the post-partum uterus, we crossed the previously described Prdm1Cre [ERT2/+] deleter strain[28] to Prdm1CA/CA mice[27] and the resulting Prdm1Cre [ERT2/CA] females injected i.p. at PP day 0 with a single dose of Tx (4 mg/25 g body weight; 156738, MP Biomedicals) diluted in corn oil (C8267-500ML, Sigma). For fate mapping experiments, Prdm1Cre [ERT2/+28] females were crossed with homozygous Rosa26R-YFP (referred to as R26R[YFP30],) or homozygous Rosa26R-lacZ (referred to as R26R[lacZ31],) males. Prdm1Cre [ERT2/+];R26R [YFP/+] or Prdm1Cre [ERT2/+];R26R [lacZ/+] females were injected i.p. with a single dose of Tx (4 mg/25 g body weight) at PP day 0 or day 2. Tissues were collected and processed for cryosectioning.

### Histology and immunohistochemistry

Uterine horns were collected, pinned into paraffin wax coated petri dishes, fixed overnight in 4% paraformaldehyde (PFA) in phosphate buffered saline (PBS, 14190144, Gibco), washed, and dehydrated using a standard ethanol series. Intact uterine horns or individual decidual swellings cleared in Histoclear (HS-202, National Diagnostics), were embedded in paraffin wax and sectioned (6 μm). Hematoxylin and eosin staining (H-3502, Vector Laboratories) was performed using standard protocols. Dewaxed sections were subject to antigen retrieval by boiling in Tris/EDTA (pH 9.0) or Citrate buffer for 1 h and permeabilised in 0.1% TritonX-100 in PBS for 10 min at RT. Antibodies and staining conditions used for detecting Blimp1, cytokeratin, pSmad1/5/9 and Mmp10 are listed in Supplementary Table 1. Sections were lightly counterstained with hematoxylin (H-3401, Vector Laboratories), coverslipped and imaged.

### Immunofluorescence and LacZ staining

PP day 5 uteri were harvested and fixed in 1% PFA/PBS overnight at 4 °C. Samples were placed in a 30% sucrose/PBS solution and then embedded in OCT (361603E, VWR chemicals) before cryosectioning (10 μm) on a Leica CM3050 S Research Cryostat. Sections were processed for IF staining, as described[9]. Samples were imaged on an Olympus Fluoview FV1000 microscope and images were processed using ImageJ software. Antibodies used are listed in Supplementary Table 1. LacZ staining from the Prdm1Cre [ERT2/+];R26R [lacZ/+] uteri cryosections was performed according to standard protocols[49]. Sections were counterstained with nuclear fast red (H-3403, Vector), mounted and imaged.

### RNA-seq and data analysis

Total RNA from wild type (Prdm1CA/+) and Blimp1 mutant decidua, (E6.5) was isolated 14 h post injection with either RU486 or mineral oil (control samples), using a RNeasy Mini kit (74104, Qiagen) with on column DNase treatment according to the manufacturer's protocol. Total RNA was ribodepleted using Ribozero and libraries prepared using standard stranded Illumina RNA-seq protocols. Paired end sequencing (150 bp) was performed on an Illumina Novaseq 6000 or NextSeq 500.

Raw sequencing reads and normalised read counts are available at NCBI GEO under accession number GSE249112.

Paired-end sequencing reads (150 bp) were mapped to the mm10 mouse genome using RNA-STAR in Galaxy (https://usegalaxy.org). Aligned BAM files were then analysed using the RNA-Seq quantitation pipeline in SeqMonk (V1.45.4). Differentially expressed genes were identified using DeSeq2 with a FDR cut-off of 0.05, >2-fold change in expression and FPKM of >1 (in all samples within at least one group). GO analysis was performed using Metascape[50] (www.metascape.org)

Gene set enrichment analysis (GSEA) was performed using all RNA-Seq genes ranked by directional fold change and DESeq2 FDR (directional fold x -log$_{10}$FDR) from most significantly up-regulated to most significantly down-regulated following RU-486-treatment, and compared with genes displaying increased (LEE_BMP2_TARGETS_UP) and decreased expression (LEE_BMP2_TARGETS_UP) in Bmp2 null uterus, 24 h after decidual trauma MsigDB (https://www.gsea-msigdb.org/gsea/msigdb/mouse/genesets.jsp)[20,22,51].

### Statistics and Reproducibility

For each experiment at least three biological replicates were used per genotype. Exact *n* is stated for every experiment in figure legends. All statistical analyses were performed using GraphPad Prism (10.2.1). Student's unpaired two tailed *t*-test was used for analysing Blimp1 expression in PP day 2 control and mutant uteri. $P < 0.05$ was considered statistically significant. Sample sizes were selected based on previous experiments and no statistical method was applied to predetermine sample size. No data were excluded from the analyses. As the phenotype of the controlled versus experimental tissue was morphologically evident, blinding was not required during experiments and outcome assessment.

### Reporting summary

Further information on research design is available in the Nature Portfolio Reporting Summary linked to this article.

## Data availability

The RNA-seq data generated in this study have been deposited in the NCBI GEO database under accession code GSE249112. The source data underlying Supplementary Figs. 4b and 5c are provided as a Source Data File. Source data are provided with this paper.

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

## Acknowledgements

We thank the Oxford Genomics Centre at the Wellcome Centre for Human Genetics (funded by Wellcome Trust grant reference 203141/Z/16/Z) for the generation and initial processing of the sequencing data, and Ita Costello for helpful discussions and technical advice. This work was supported by a grant from the Wellcome Trust (214175/Z/18/Z). E.J.R. is a Wellcome Trust Principal Fellow.

## Author contributions

M-E. X., M.G., E.J.R. and A.W.M. designed the experiments. M-E. X., M.G., E.J.R. and A.W.M. performed the experiments. M-E. X., E.K.B., E.J.R. and A.W.M. contributed to writing the paper.

## Competing interests

The authors declare no competing interests.
