## [Peer Review file · Nature Communications]

The zinc-finger transcription factor Blimp1/Prdm1 is required for uterine remodelling and repair in the mouse

Corresponding Author: Professor Elizabeth J. Robertson

Version 0:

Reviewer comments:

Reviewer #1

(Remarks to the Author)

Postpartum uterine involution is essential for successful subsequent pregnancies. However, the mechanisms of uterine remodeling and repair after parturition are still poorly understood. This study presents evidence that Blimp1-deficient mice undergo abrupt uterine repair during normal parturition or RU486-induced pregnancy termination. While the disrupted expression of MMPs and BMP2 may contribute to this phenotype, this study is too descriptive and suffers from shallow mechanism and lacks in-depth mechanistic analysis to adequately support the hypothesis that Blimp1 is necessary for tissue remodeling.

Major comments are listed below:

1 The results from Fig.1 to Fig.5 only partially describe the phenotypes of the abrupted tissue remodeling in Blimp1 deficient mice. Some more evidence is required to thoroughly estimate the process of tissue remodeling during early pregnancy and post-parturition.

2 Blimp1 regulated genes in tissue remodeling are investigated by RNA-Seq in RU486 treated mice (Fig 6). MMPs and BMP2 are picked up and confirmed by IHC (Fig 7). But the physiological significance and the necessity and causal relationship of these genes in tissue modeling in pregnancy, especially their role in post-parturitional tissue repairing remain elusive. More convincing genetic and biochemical evidence are required.

3 How Blimp1 regulates these target genes require more evidence. Whether Blimp1 regulate these genes directly or through other pathways need to be efficiently illustrated.

4 The relationship between Blimp1 and ER and PR also need to be fully estimated. Based on the results of RNA-Seq, it appears that, PR response is aberrant in Blimp1 knockout mice, but the underlying mechanism is rarely studied in current study.

5 In Fig 1C, it appears that there are plenty of glands in both M and AM sites. Please make sure this is the right orientation of uterus. Cross-section and 3D staining is required to confirm this result.

Reviewer #2

(Remarks to the Author)

The submitted manuscript showed a very interesting phenotype in Blimp1 mutant females where uterine tissue resolution and following repair do not occur. The suggested theme by the authors is very attractive and observed phenotypes of Blimp1 mutant females are intriguing. However, it is difficult to find the definitive function of Blimp1 in this process since the data in the manuscript is somewhat descriptive, mainly based on immunostaining. Furthermore, although the authors tried to explain the molecular function of Blimp1 during this process with bioinformatic analyses for expression profiles between wildtype and mutant uterine samples, the suggested hypothesis seems to be also descriptive and could be one of the possibilities

that come from the Blimp1 deficiency in the uterus. To demonstrate the definitive function of Blimp1 during this process, the manuscript should have demonstrated how Blimp1 participates in uterine tissue clearance followed by tissue repair at the molecular levels as well. Detailed comments are listed below

1. Innate immunity is critical for the clearance of damaged tissues followed by tissue repair. The inability to resolve decidual remnants in mutant females could be derived from dysfunction of immune cells which are mainly involved in these processes for tissue regeneration after injury and/or inflammation. The authors also pointed it in lines 169-172 "By contrast 14 hours post RU486 treatment when the embryo has disintegrated, and the deciduum lacking its normal compact cellular architecture has become infiltrated by maternal blood cells, the pattern of Blimp1 expression dramatically changes". However, the authors did not pursue whether infiltration of immune cells is dysregulated in Blimp1 mutant uterus. It is very important to pursue how infiltrated immune cells participate in the clearance of disintegrated uterine tissues and following repair processes.

2. In along with the above comment, in a previous work (Nature Communications in 2020) published by the authors, they demonstrated a significant increase in the spread of macrophages (Supplementary Fig. 3b) as well as increased total macrophage numbers (Supplementary Fig. 3c) around the demising embryos in pregnant Blimp1 mutant females. They also strongly suggested that Blimp1 normally silences Csf1 expression to prevent macrophage invasion into the implantation site. Thus, the authors are supposed to follow how immune networks are involved in sequential events of tissue repair following tissue clearance in the uterus using Blimp1 mutant mice in this manuscript as well.

3. It is very important to find out how long disintegrated remnants remain in the Blimp1 mutant uterus and whether Blimp1 mutant females could get pregnant again. If the disintegrated remnants are finally cleared, it is critical to understand how this process occurs at that time in these mutant mice. These data are strongly required to understand the definitive function of Blimp1 in the uterus during tissue clearance.

4. The authors used two different models, pregnancy (Figures 1 and 2), and RU486 treatment (Figures 3 to 5), to point out the necessity of Blimp1 during tissue clearance of a disintegrated uterus. It is very important to support the requirement of Blimp1 during this process with two independent model systems. However, all data in Figures 1 to 5 are immunostaining-based observations. Whereas the authors tried to explain the molecular mechanism of Blimp1 during this process in Figures 6 and 7, these figures also simply showed a summary of RNA sequencing with a hypothesis that could come from previous works performed by the authors and others. In the abstract, the authors tried to suggest that "the failure to up-regulate the expression of the matrix metalloprotease Mmp10 in combination with insufficient suppression of BMP signalling, likely explain Blimp1-dependent phenotypic changes. Overall these results identify Blimp1 as a master regulator of uterine tissue remodelling and repair". This suggestion is very attractive, but again descriptive with a well-designed hypothesis and one of various possibilities from the datasets shown in this manuscript. Authors are strongly recommended to perform any mechanistic experiments to directly demonstrate how Blimp1 participates in this process, such as experiments to rescue observed phenotypes.

5. In Figure 2a, the authors mentioned that Blimp1 is expressed in stromal cells and regenerating luminal epithelium at PP2 at sites of placental detachment. If they would like to say so, co-immunostaining of Blimp1 and Pan-cytokeratin is strongly recommended to make sure that they are colocalized at this site. In addition, Blimp1 is found in the newly forming epithelium in Figure 4. However, there is no direct evidence that Blimp1-expressing cells are newly forming epithelium. Considering that this is a very important scientific observation, the authors should have provided more definitive data to support their suggestion, such as co-immunostaining between Blimp1, pan-cytokeratin, and Ki67 co-staining.

6. In Fig 3b, enlarged images of upper boxes for control and RU486-treated uterine tissue do not seem to be in the same area. In addition, considering that embryos may not be completely demised 14h after RU486 treatment, it is better to take images with embryos not only in control but also in the RU486 treatment group.

7. It is not clear why the authors performed experiments at different days of pregnancy such as E6.5 in Fig 5a and 72h after RU486 at E5.5 in Fig 5b and 5c.

Version 1:

Reviewer comments:

Reviewer #1

(Remarks to the Author)

Although the author provides additional experiments, the key question still remains unclear about the molecular mechanism of how Blimp1 participates in tissue remodeling. Although DEGs were screened by RNA-Seq in WT and KO mice, the direct downstream target genes of Blimp1 remain elusive. How Blimp1 regulates these direct target genes also requires deeper evidence. The application of Blimp1Cre-ERT2 mice also only proves the essential role of Blimp1 in tissue repairment. But the convincing evidence of the underlying mechanism remains untouched.

For concern 1: Too limited markers are applied to estimate the phenotype of Blimp1 deficient uterus.

For concern 2: The in vivo role of MMP10 and BMP2 in uterine repairment remains unclear.

For concern 3: The regulation apparatus of BLIMP1 in the uterus is still largely unknown.

In a word, this investigation provides limited information in the field of reproduction.

Reviewer #2

(Remarks to the Author)

As I mentioned in the reviewer's comment, the submitted manuscript has interesting observations and the authors suggested an attractive hypothesis on the phenotypes of Blimp1 deficient uteri. I absolutely agree with the authors on the fact that uterine tissue resolution and following repair do not occur in Blimp1 mutant mice including a new model of Blimp1 deficient condition. However, the authors did not provide direct evidence to support the main theme of the submitted manuscript because of a lack of mechanistic approaches to demonstrate the direct role of Blimp1 in this process at the molecular level even in the revised manuscript. Although the authors included new experiments with additional genetic models for new Figures, such as Figures 8 and 9, they provide descriptive information again without any mechanistic approaches. Furthermore, it is now confusing since Figure 9 provides evidence that Blimp1 positive cells are restricted to a subset of stromal cells and not transformed to be included in the newly formed epithelium. The authors suggested the indirect role of Blimp1 expressed only in the stroma for epithelial regeneration in Figure 9. However, the initially submitted manuscript focused on the direct role of Blimp1 in epithelial repopulation during the repair process. Although the phenotypes of Blimp1 deficient females are quite interesting and the authors suggested an interesting hypothesis, the revised manuscript does not contain data to demonstrate the direct role of Blimp1 on these events at the molecular level.

1. Author response to comments 1 and 2

Since a variety of immune cells are positive for ER and/or PR and they are necessary for tissue clearance and following repair processes, it is critical to examine how infiltrated immune cells participate in these events as mentioned by the reviewer (comments 1 and 2 of reviewer 2). However, the authors considered this comment as "not relevant to this study" and did not pursue both comments related to this issue in the revised manuscript.

2. Author response to comment 3

I think that the authors responded well to this comment and revised the manuscript accordingly.

3. Author response to the comment 4

The authors brought another genetic approach to answer comment 4. I mentioned above, I have no doubt about the evidence that timely post-partum tissue repair does not occur in the conditions that Blimp1 is deleted (including a conditional deletion of Blimp1). However, new Figures included in the revised manuscript (Figures 8 and 9) fail to demonstrate any direct role of Blimp1 in this process. These immunostaining-based data provide indirect and descriptive information again. Furthermore, the authors insist that Mmp10 and Blimp1 are co-expressed in the PP day 2 uterus in Figure 7. However, the results of IHC for Mmp10 and Blimp1 are not good enough to support what the authors mentioned in the revised manuscript and the quality of images is not convincing.

4. Author response to the comment 5.

It is quite understandable to have technical difficulties in performing co-immunofluorescence since available antibodies are limited. Although the authors mentioned that they performed CK and Blimp1 immunostaining in serial sections to demonstrate their colocalization, the revised manuscript does not have images to meet the comment. The authors also mentioned that Figure 9 with the fate mapping experiment could answer this comment. However, I am not sure how Figure 9 provided data to respond to comment 5. Figure 9 describes that Blimp1 expression is restricted to a subset of stromal cells underlying luminal epithelium, but does not give any further information.

5. Author response to the comment 6

Although the embryo may not be included in RU-486 treated group due to the reason that the authors suggested, it is doable to have a histologically matched area between the control and RU-486 treated groups as suggested by the reviewer. However, the authors did not pursue any further experiments to show better images to meet the comments in the revised manuscript.

6. Author response to the comment 7

I think that the authors responded well to this comment.

Version 2:

Reviewer comments:

Reviewer #1

(Remarks to the Author)

Although the author finds an interesting phenotype of epithelium repair, there are still tons of concerns need to be carefully addressed.

1 In this revised manuscript, minimal progress has been made in addressing my previous concerns. The study remains largely descriptive, with limited evidence to support the causal relationship between Blimp1 and the differentially expressed

genes (DEGs) identified through RNA-Seq. To confirm the direct regulatory role of BLIMP1 on its target genes, high-quality BLIMP1 ChIP-Seq or CUT&RUN data are essential. These methods are well-established; therefore, the inability to perform these experiments in the author's lab should not serve as an excuse to avoid exploring the molecular mechanisms of BLIMP1. Additionally, the rationale to infer the BLIMP1 regulated gene in epithelium regeneration using RU486 treated uterus is also questionable.

2 There is considerable crosstalk between the stroma and epithelium. While the author employs a CreERT mouse model to demonstrate that the regenerated epithelium primarily not originates from the stroma, the role of the stroma, especially the paracrine factors derived from stroma, in re-epithelialization remains unclear. The mechanisms by which BLIMP1 and MMP10 in the stroma contribute to this process are not well understood. More compelling in vivo evidence is needed to confirm the critical role of MMP10 in re-epithelialization.

3 Additionally, since BLIMP1 is expressed in both the stroma and epithelium, the specific contribution of each compartment to re-epithelialization remains unclear. The role of epithelial BLIMP1 in this process should also be carefully evaluated.

4 The author suggests that BLIMP1's function is well-studied in other tissues. However, given the unique context of the uterus, it is challenging to assume that the regulatory mechanisms of BLIMP1 are the same in this organ.

5 The ability of cells to proliferate should be assessed.

6 Previous studies suggest that glandular AXIN2+ cells may act as potential epithelial progenitors. The relationship between BLIMP1 and AXIN2 should also be explored.

Reviewer #2

(Remarks to the Author)

The authors have tried to answer the comments from reviewers in the revised manuscript. The authors have performed a few experiments to change low-profile images to meet the reviewers' comments. As mentioned in 1st review, the manuscript showed interesting phenotypes of Blimp1 deficient uterus during uterine remodeling and repair. However, the main concern for the manuscript is that the revised manuscript does not contain any direct evidence of how Blimp1 contributes to uterine re-epithelialization during tissue remodeling and repair. The authors kept saying that the reviewers did not provide specific suggestions to answer the comments. However, if the manuscript has mechanistic approaches to demonstrate the answer to the main concern, the reviewers could provide detailed experimental suggestions to support and strengthen what the authors performed. I am afraid that the revised manuscript has no mechanistic approaches to directly demonstrate what Blimp1 exactly does in the stromal cells adjacent to proliferating new luminal epithelium during uterine repair. Furthermore, the manuscript focused on Bmp2 signaling and Mmp2 expression during this process, but they do not have a direct link to data in Figure 9. Collectively, the revised manuscript shows interesting phenotypes of Blimp1 deficient uterus during the repair process, with dysregulation of various signaling pathways, including Bmp2 and Mmp2, but is a descriptive study without direct evidence of Blimp1 function during uterine repair.

RESPONSES TO THE REVIEWERS COMMENTS

Both reviewers expressed concerns that we lack sufficient evidence that Blimp1 is required for uterine remodelling and repair.

In the original manuscript we reported that loss of Blimp1 within the PR+ cell population of the mouse uterus prevents decidual shedding during pregnancy with decidual tissue persisting at the equivalent of 4 weeks post-partum stages. Moreover administration of the abortifacient RU486 fails to cause release of decidual swellings in the absence of Blimp1 activity.

Here, in the revised manuscript, to unequivocally establish a role for Blimp1 in repair during post-partum uterine involution we exploited the Prdm1-CreERT2 allele (knock-in of a CreERT2 cassette into Blimp1), in combination with the floxed Prdm1/Blimp1 allele to generate Blimp1-CreERT2/Blimp1^{CA} females. In this genetic context a single dose of tamoxifen administered on PP day 0 conditionally deletes Blimp1 specifically within the Blimp1⁺ uterine stromal population. The resulting loss of >85% of Blimp1 immunoreactive cells at PP day 2 (New Fig. 8) significantly impairs the ability of sites of placental detachment to re-epithelize. Moreover the process of involution, normally completed by PP day 5, is significantly delayed and by PP day 7 the surface area of the uterine lumen has failed to reduce and remodel and rather remains highly folded (Fig 8).

Moreover, we now include new data showing that Mmp10 is normally co-expressed with Blimp1 in the placental detachment sites and uterine stroma at PP day 2 (Fig. 7) and that Mmp10 fails to be induced following Blimp1 conditional deletion (Fig. 8). Collectively these new data conclusively demonstrate the essential role of Blimp1 in the post-partum uterus in induction of this important extra-cellular remodelling enzyme.

Additionally we have now performed genetic fate mapping experiments to test whether the Blimp1⁺ stromal cells represent a population of multipotent stem cells. We generated females carrying both the Blimp1CreERT2 allele and a Cre-activatable constitutive reporter (ROSA26RYFP or ROSA26RLacZ) and pulse labelled the Blimp1⁺ stromal cells at PP day2. We conclusively demonstrate that the population of Blimp1⁺ stromal cells do not directly contribute to the regenerating epithelium at any position along the involuting uterus. Rather Blimp1 regulates a gene network within stromal cells that have essential roles in promoting the rapid tissue repair and re-modelling process. This new fate mapping data is shown in Figure 9.

Both Reviewers suggested that we try to provide “mechanistic insights” into how Blimp1 controls tissue repair and re-modelling. We have now carried out ATAC-seq experiments comparing chromatin accessibility in wild type versus PRCre deleted decidual tissue (same tissues used for the RNA-seq experiments). Unfortunately, despite considerable investment of time and money, this analysis failed to provide clear insights into the underlying mechanism. We discovered in the absence of Blimp1 activity that numerous genes normally regulated by both PR and ER signalling loose chromatin accessibility (please see the attached file summarising the data). Unfortunately however we were unable to make correlations with Blimp1 occupancy. This is probably because we had to perform ATAC-seq and Blimp1-ChIP-seq using heterogeneous bulk tissue samples. To accurately identify direct/indirect Blimp1 targets it would be necessary to repeat these experiments using purified populations of Blimp1⁺ wild type and the corresponding Blimp1 deleted cell populations from the decidual tissues. Unfortunately this experiment is not currently technically feasible and lies outside the scope of the current manuscript.

The revised manuscript that now includes two new sets of genetic experiments (data provided in Figures 8 and 9, and Supp Figures 4-6) has been edited to include these additional Figures, and the text and figures re-ordered to present the pregnancy remodelling data (Figures 1-6) in the first half of the paper and the post-partum uterine data (Figures 7-9) at the end of manuscript.

Additional information has been also been added to the main Figures and Supp Figures in response to the Reviewers Comments (detailed below).

Responses to the Specific Reviewer points:

Reviewer #1:

Postpartum uterine involution is essential for successful subsequent pregnancies. However, the mechanisms of uterine remodelling and repair after parturition are still poorly understood. This study presents evidence that Blimp1-deficient mice undergo abrupt uterine repair during normal parturition or RU486-induced pregnancy termination.

The reviewer has mis-understood the phenotypic defects described in our paper. Blimp1 deficient mice are not undergoing “abrupt uterine repair”. Rather Blimp1 deficient mice entirely **fail** to maintain a viable pregnancy and thus do not undergo normal parturition. The decidual cell masses formed at day 5.5 of pregnancy (which lack viable embryos) **fail** to be shed from the uterus, become encapsulated by epithelium and persist until at least the equivalent of PP day 28. In addition RU486 administration **fails** to induce pregnancy termination, rather decidual shedding is entirely refractory to RU486 treatment.

While the disrupted expression of MMPs and BMP2 may contribute to this phenotype, this study is too descriptive and suffers from shallow mechanism and lacks in-depth mechanistic analysis to adequately support the hypothesis that Blimp1 is necessary for tissue remodelling.

Our experiments that demonstrate Blimp1 requirements during RU486 mediated decidual shedding and repair during gestation, and in the post-partum repair process were admittedly very descriptive. We have now exploited a Blimp1Cre-ERT2 allele (Elias et al 2017) in conjunction with a Blimp1-floxed allele (Shapiro-Shalef et al, 2003) to eliminate Blimp1 function in the immediate post-partum uterus by a single administration of tamoxifen. We directly demonstrate that robust depletion of immunoreactive Blimp1 protein is associated with a significant impairment of the repair process at the sites of placental detachment. At PP day 2 the CK+ epithelial layer fails to regenerate and significant amounts of blood and tissue debris are found within the uterine lumen. Moreover, in the revised manuscript we now demonstrate that Mmp10 is normally upregulated within the Blimp1 domain at PP day 2 and establish the direct connection between Mmp10 and Blimp1, since loss of Blimp1 similarly results in failure to induce Mmp10 expression during post-partum repair.

Major comments are listed below:

1 The results from Fig.1 to Fig.5 only partially describe the phenotypes of the abrupted tissue remodeling in Blimp1 deficient mice. Some more evidence is required to thoroughly estimate the process of tissue remodeling during early pregnancy and post-parturition.

New data in Figure 8 and Supplementary Fig 5 describe the mutant phenotype and impact on tissue repair and involution resulting from conditional removal of Blimp1 from the post-partum uterine stroma. These new experiments provide unambiguous evidence that Blimp1 expression in the uterine stroma contributes to both re-epithelialization at the sites of placental detachment as well as remodelling of the luminal epithelium along the entire length of the uterine horns. As shown in new Figure 8 endometrial involution, normally achieved by PP day 5 is severely impaired, and remarkably the uterine epithelium remains highly folded at PP day 7.

2 Blimp1 regulated genes in tissue remodeling are investigated by RNA-Seq in RU486 treated mice (Fig 6). MMPs and BMP2 are picked up and confirmed by IHC (Fig 7). But the physiological significance and the necessity and causal relationship of these genes in tissue modeling in pregnancy, especially their role in post-parturitional tissue repairing remain elusive. More convincing genetic and biochemical evidence are required.

In the revised manuscript we further explore the relationship between Blimp1 and Mmp10 in the post-partum uterus. Our new experiments clearly demonstrate that ablation of Blimp1 in the PP uterus results failure to induce Mmp10 expression at the sites of placental detachment at PP day 2.

3 How Blimp1 regulates these target genes require more evidence. Whether Blimp1 regulate these genes directly or through other pathways need to be efficiently illustrated.

An extensive literature has documented the multiple roles that Blimp1 plays in numerous different tissue contexts ranging from primordial germ cells, fetal intestinal epithelium, immune cell sub-sets, placenta etc. A major goal over the past decades has been to identify Blimp1 regulated target genes in these diverse tissue settings. In an effort to learn more about Blimp1 targets in the uterus we carried out ATAC-seq experiments. We compared chromatin accessibility in wild type versus PRCre deleted decidual tissue (same tissues used for the RNA-seq experiments). Unfortunately this analysis failed to provide meaningful insights with regard to Blimp1 mechanism of action. Rather gene set enrichment analysis shows that loss of Blimp1 correlates with loss of accessibility at a sub-set of previously described PR and ER responsive genes (please see the attached file summarising our data for the reviewers information). For technical reasons we can only perform genomic experiments (ATAC-seq, ChIP-seq etc) in bulk heterogeneous tissue in which only a small sub-set of cells express Blimp1. To reliably identify Blimp1 regulated genes and the associated downstream gene networks it would be necessary to isolate the Blimp1+ive and corresponding Blimp1 deleted uterine cell populations – unfortunately this is not technically feasible at present.

4 The relationship between Blimp1 and ER and PR also need to be fully estimated. Based on the results of RNA-Seq, it appears that, PR response is aberrant in Blimp1 knockout mice, but the underlying mechanism is rarely studied in current study.

The ATAC-seq analysis (appended) provides evidence loss of chromatin accessibility at a sub-set of PR and ER target genes caused by lack of Blimp1. It is intriguing that Blimp1-dependent up-regulated Mmp10 decidual expression is caused by RU486 administration, and that Mmp10 regulation in the wild type post-partum uterus correlates with the drop in the level of circulating progesterone levels at parturition. We hope to learn more about the underlying mechanism in future experiments.

5 In Fig 1C, it appears that there are plenty of glands in both M and AM sites. Please make sure this is the right orientation of uterus. Cross-section and 3D staining is required to confirm this result.

The presence of decidual remnants causes disturbances in the orientation of the horns as reflected in the positioning of the glands – this is likely due to the swollen fluid filled regions between the decidual sites. This phenomenon is also seen in the sections presented in Supp Figs 1&2. In all experiments to preserve the A-AM orientation samples are pinned prior to fixation and processing. However, the CK staining simply serves to demarcate the uterine epithelium and the epithelial population that invests the residual decidual swellings.

Reviewer #2:

The submitted manuscript showed a very interesting phenotype in Blimp1 mutant females where uterine tissue resolution and following repair do not occur. The suggested theme by the authors is very attractive and observed phenotypes of Blimp1 mutant females are intriguing. However, it is difficult to find the definitive function of Blimp1 in this process since the data in the manuscript is somewhat descriptive, mainly based on immunostaining. Furthermore, although the authors tried to explain the molecular function of Blimp1 during this process with bioinformatic analyses for expression profiles between wildtype and mutant uterine samples, the suggested hypothesis

seems to be also descriptive and could be one of the possibilities that come from the Blimp1 deficiency in the uterus. To demonstrate the definitive function of Blimp1 during this process, the manuscript should have demonstrated how Blimp1 participates in uterine tissue clearance followed by tissue repair at the molecular levels as well. Detailed comments are listed below

1. Innate immunity is critical for the clearance of damaged tissues followed by tissue repair. The inability to resolve decidual remnants in mutant females could be derived from dysfunction of immune cells which are mainly involved in these processes for tissue regeneration after injury and/or inflammation. The authors also pointed it in lines 169-172 "By contrast 14 hours post RU486 treatment when the embryo has disintegrated, and the deciduum lacking its normal compact cellular architecture has become infiltrated by maternal blood cells, the pattern of Blimp1 expression dramatically changes". However, the authors did not pursue whether infiltration of immune cells is dysregulated in Blimp1 mutant uterus. It is very important to pursue how infiltrated immune cells participate in the clearance of disintegrated uterine tissues and following repair processes.

2. In along with the above comment, in a previous work (Nature Communications in 2020) published by the authors, they demonstrated a significant increase in the spread of macrophages (Supplementary Fig. 3b) as well as increased total macrophage numbers (Supplementary Fig. 3c) around the demising embryos in pregnant Blimp1 mutant females. They also strongly suggested that Blimp1 normally silences Csf1 expression to prevent macrophage invasion into the implantation site. Thus, the authors are supposed to follow how immune networks are involved in sequential events of tissue repair following tissue clearance in the uterus using Blimp1 mutant mice in this manuscript as well.

The role of immune cell infiltration in normal tissue clearance and repair has been extensively studied in a variety of contexts. The current study does not concern the Blimp1-dependent immune network involved in normal tissue repair. As for all sites of tissue and wound repair, the involvement of infiltrating immune cells is obviously of key importance and occurs in response to factors released by damaged tissues. However, as the "damage and subsequent repair" does not occur in the Blimp1 depleted uterus during pregnancy this is not relevant to the current study.

3. It is very important to find out how long disintegrated remnants remain in the Blimp1 mutant uterus and whether Blimp1 mutant females could get pregnant again. If the disintegrated remnants are finally cleared, it is critical to understand how this process occurs at that time in these mutant mice. These data are strongly required to understand the definitive function of Blimp1 in the uterus during tissue clearance.

In the revised manuscript we examined the appearance of Blimp1 depleted uterine horns at the equivalent of 28 PP days (i.e. 7 weeks after copulation plugs were detected) in Supp Figure 2 b. The epithelial invested decidual remnants still persist and block the uterine lumen. We find no evidence that the uteri are repaired. The females cycle normally as seen by the accompanying ovarian histology, but released oocytes cannot be fertilized in the oviducts as the lumen is fully occluded and hence the females cannot get pregnant.

4. The authors used two different models, pregnancy (Figures 1 and 2), and RU486 treatment (Figures 3 to 5), to point out the necessity of Blimp1 during tissue clearance of a disintegrated uterus. It is very important to support the requirement of Blimp1 during this process with two independent model systems. However, all data in Figures 1 to 5 are immunostaining-based observations. Whereas the authors tried to explain the molecular mechanism of Blimp1 during this process in Figures 6 and 7, these figures also simply showed a summary of RNA sequencing with a hypothesis that could come from previous works performed by the authors and others. In the abstract, the authors tried to suggest that "the failure to up-regulate the expression of the matrix metalloprotease Mmp10 in combination with insufficient suppression of

BMP signalling, likely explain Blimp1-dependent phenotypic changes. Overall these results identify Blimp1 as a master regulator of uterine tissue remodelling and repair". This suggestion is very attractive, but again descriptive with a well-designed hypothesis and one of various possibilities from the datasets shown in this manuscript. Authors are strongly recommended to perform any mechanistic experiments to directly demonstrate how Blimp1 participates in this process, such as experiments to rescue observed phenotypes.

We thank the reviewer for this suggestion. While it is not technically possible to restore Blimp1 function within the PR uterine tissues and "rescue" the phenotype, we have now performed the reciprocal experiment. We have conditionally removed Blimp1 activity from within the expression domain in the immediate post-partum uterus. This genetic manipulation provides conclusive evidence that Blimp1 is required for efficient and timely post-partum tissue repair. We also find that the process of epithelial remodelling along the length of the post-partum uterus that accompanies involution is severely impaired, and the tissue retains a highly folded appearance at PP day 7 (Figure 8). Moreover we demonstrate that Mmp10 is normally robustly expressed in the PP day 2 uterus at sites coincident with Blimp1 expression (Figure 7) and removal of Blimp1 activity from the stroma prevents up-regulation of Mmp10 expression (new Figure 8). We have also performed fate mapping experiments to demonstrate that the Blimp1+ cells present within the stroma do not represent a pool of "stem cells" that directly contribute to the uterine epithelial layer (new Figure 9). Rather Blimp1 activates a gene network within the stroma that is responsible for promoting repair and remodelling of the adjacent epithelial layer.

5. In Figure 2a, the authors mentioned that Blimp1 is expressed in stromal cells and regenerating luminal epithelium at PP2 at sites of placental detachment. If they would like to say so, co-immunostaining of Blimp1 and Pan-cytokeratin is strongly recommended to make sure that they are colocalized at this site. In addition, Blimp1 is found in the newly forming epithelium in Figure 4. However, there is no direct evidence that Blimp1-expressing cells are newly forming epithelium. Considering that this is a very important scientific observation, the authors should have provided more definitive data to support their suggestion, such as co-immunostaining between Blimp1, pan-cytokeratin, and KI67 co-staining.

We initially performed immunofluorescence staining. However because the CK staining is so bright it is difficult to accurately assess whether any of the Blimp1+ cells co-express CK by IF. For this reason that we present serial sections stained for CK and Blimp1 respectively. We have also stained for KI67+ cells but, similar to the report by Huang et al 2012, only a few positive cells are detectable and those present are confined to the stroma. However the new fate mapping experiments shown in new Figure 9 using two different reporter alleles (GFP and LacZ) unambiguously demonstrate that the Blimp1+ stromal cells, while closely intermingled and intercalating with the regenerating CK+ epithelial cells at detachment sites, do not ultimately contribute to the repaired epithelial layer.

6. In Fig 3b, enlarged images of upper boxes for control and RU486-treated uterine tissue do not seem to be in the same area. In addition, considering that embryos may not be completely demised 14h after RU486 treatment, it is better to take images with embryos not only in control but also in the RU486 treatment group.

At this later stage the onset and demise of the embryos occurs very rapidly in response to RU486, and the deciduae collapse and expel the embryonic tissue, making it almost impossible to histologically capture the dying embryos per se – the RU486 sample images in Fig 3b are taken at the midpoint of the deciduum do in fact show fragments of the embryo including the yolk sac (now indicated in the images). These sections are not intended to show the appearance of the embryo proper but rather highlight the marked differences in the distribution of Blimp1 expressing cell populations present in the surrounding **maternal tissues**.

7. It is not clear why the authors performed experiments at different days of pregnancy such as E6.5 in Fig 5a and 72h after RU486 at E5.5 in Fig 5b and 5c.

To highlight the profound differences between the wild type and mutant uteri we performed the experiments at different stages. The experiment in Fig 5a shows the histological appearance of wild type versus Blimp1 deficient deciduae 14 hrs post RU486 treatment at the equivalent of day 6. The lower panels in Figure 5 extend the observation to emphasise that the mutants are completely refractory to RU486. Decidual release is not just delayed, rather the decidual tissue persists and is histologically normal some 72 hours post RU486 administration (Fig 5c).

ATAC-seq experiments addressing the effects of Blimp1 loss on chromatin accessibility in the 6.0 dpc deciduum

To elucidate why Blimp1 mutant decidua are refractile to progesterone signalling we performed ATAC-seq experiments to see if loss of Blimp1 alters chromatin accessibility. Triplicate pools of wild type and Blimp1 mutant decidua ($n > 6$ per group) were enzymatically dissociated, and the live cells sorted by FACS by gating on 7AAD negative cells. ATAC-seq libraries were generated using custom Nextera primers, sequenced on one lane of a HiSeq4000 (75bp paired end) and mapped to the mouse genome (mm10).

After removal of duplicate reads, ATAC-seq peaks in each sample were identified using Genrich. Peak coordinates for all samples were pooled and the overlapping regions collapsed to generate a peak region annotation file. BAM read files for each sample and the peak regions annotation file were then imported into Seqmonk (V1.45.14). Read count quantitation was performed correcting for total read count to largest store log transformed by enrichment normalisation quantitation from percentiles 20.0 to 90.0. Comparison of read counts in ATAC-seq peak regions between genotypes was performed using Limma Stats. For gene set enrichment analysis, changes in chromatin accessibility in ATAC-seq peak regions were ranked from most increased to most decreased in mutants relative to wild types (based on direction fold change $\times -\log_{10}$ FDR). Ranked peak regions were compared with overlapping uterine progesterone receptor (Rubel et al., 2012) and oestrogen receptor (Hewitt et al., 2012) ChIP-seq peak coordinates to determine if alterations in chromatin accessibility in Blimp1 mutants correlates with progesterone or oestrogen receptor binding sites. Leading edge ATAC-seq peak regions driving statistical associations in each gene set enrichment analysis were imported into GREAT to identify associated genes using the default basal plus extension rule (+5kb to -1 of annotated TSS, plus distal to 1000 kb).

Results

Chromatin accessibility at previously identified uterine progesterone and oestrogen receptor binding sites was found to be significantly reduced in Blimp1 mutant decidua. These data are consistent with our previous findings that pregnancy hormone-driven decidual growth is impaired in Blimp1 mutants. While these data may partially explain why the mutant deciduae are refractile to the effects of treatment with the progesterone receptor antagonist RU486 in the present study, interpretation of the results in this context is clouded by the spatial and temporal changes in Blimp1 signalling in the decidua upon RU486 treatment that are evident from the immunostaining experiments. A single cell ATAC-seq approach may be more mechanistically informative but would take a considerable amount of time and expense. Since the present ATAC-seq data does not offer a significant insight into the mechanism underlying the resistance of blimp mutants to RU486 treatment we have decided not to add these results to the revised manuscript.

References

Rubel et al., (2012). Research resource: Genome-wide profiling of progesterone receptor binding in the mouse uterus. *Mol. Endocrinol.* 26, 1428-42.

Hewitt et al. (2012). Research resource: whole-genome estrogen receptor α binding in mouse uterine tissue revealed by ChIP-seq. *Mol Endocrinol.* 26, 887-98

Chromatin accessibility at progesterone receptor binding sites is significantly decreased in mutant

Gene set enrichment analysis

Dataset	2ATAC ranked list.rnk
Phenotype	NoPhenotypeAvailable
Upregulated in class	na_neg
GeneSet	PR_CHIP
Enrichment Score (ES)	-0.6810538
Normalized Enrichment Score (NES)	-2.4559996
Nominal p-value	0.0
FDR q-value	0.0
FWER p-Value	0.0

Gene ontology analysis of leading- edge peak regions using GREAT

Chromatin accessibility at oestrogen receptor binding sites is significantly decreased in mutant

Gene set enrichment analysis

Dataset	2ATAC ranked list.rnk
Phenotype	NoPhenotypeAvailable
Upregulated in class	na_neg
GeneSet	ER_CHIP_PEAK
Enrichment Score (ES)	-0.584193
Normalized Enrichment Score (NES)	-2.0551133
Nominal p-value	0.0
FDR q-value	0.0
FWER p-Value	0.0

Gene ontology analysis of leading- edge peak regions using GREAT

GO Biological Process

RESPONSE TO THE REVIEWER COMMENTS

Reviewer #1 (Remarks to the Author):

We were pleased to see that the Reviewer agrees that the inclusion of the BlimpCre-ERT2 post-natal data strengthens our conclusion that Blimp1 plays an essential role in uterine tissue repair.

Although the author provides additional experiments, the key question still remains unclear about the molecular mechanism of how Blimp1 participates in tissue remodeling. Although DEGs were screened by RNA-Seq in WT and KO mice, the direct downstream target genes of Blimp1 remain elusive. How Blimp1 regulates these direct target genes also requires deeper evidence. The application of Blimp1Cre-ERT2 mice also only proves the essential role of Blimp1 in tissue repairment. But the convincing evidence of the underlying mechanism remains untouched.

Neither Reviewer sent us any specific suggestions for experiments to address underlying mechanism. Moreover neither commented on our ATAC-seq experiments comparing chromatin accessibility in wild type versus PRCre deleted decidual tissue (provided in the original rebuttal), which we carried out at considerable expense (breeding a cohort of mice of the appropriate genotypes, ATAC-seq kits, sequencing costs etc). We discovered that numerous genes normally regulated by PR and ER signalling lose chromatin accessibility in the absence of Blimp1 activity. This ATAC-seq data strongly suggests that in the absence of Blimp1 that stromal cells are impaired in their ability to respond to circulating progesterone and estrogen. Experiments required to further describe “how Blimp1 regulates these direct target genes” represent long term future goals and are clearly outside the scope of the current manuscript.

For concern 1: Too limited markers are applied to estimate the phenotype of Blimp1 deficient uterus.

In the absence of specific suggestions by the Reviewer, we were unable to identify what additional markers s/he thinks would be informative to estimate the phenotype. Should we analyse Blimp1 deficient gestational stages or the post-partum uterus? In the absence of clear suggestions for specific experiments we have been unable to address this concern.

For concern 2: The in vivo role of MMP10 and BMP2 in uterine repairment remains unclear.

The role of Bmp2 signalling in the uterine stroma has been extensively examined in numerous publications (cited in our manuscript) which establish its essential roles promoting the uterine decidualization process.

As stated in our manuscript, MMP10 is not required for uterine function and homeostasis since null females are viable and fully fertile. The various members of the MMP family are highly redundant, and indeed numerous MMPs are expressed in the post-partum rodent uterus (see for example Ngoc Nguyen et al, 2016). In spite of the very large body of literature describing the individual and shared substrates of this family of extra-cellular proteases, to date deleting individual MMP family members from the mouse genome has failed to reveal unique phenotypes with respect to tissue clearance and repair.

For concern 3: The regulation apparatus of BLIMP1 in the uterus is still largely unknown.

We and others have extensively described Blimp1 mechanisms of action. Its transcription originates from a minimum of 3 promoters spaced with >80Kb 5' of the gene body (Morgan et al, 2012). However, to date the regulatory enhancers dictating its dynamic patterns of transcription in numerous tissue settings have yet to be identified. Blimp1 activities are known to be essential in numerous embryonic and adult tissue types. As for B lymphocytes, germ cells, mammary epithelium, skin etc Blimp1 in the uterus likely regulates target gene expression via its interactions with epigenetic modifiers including G9a, Prmt5, Lsd-1 etc (Bikoff et al., 2009).

In a word, this investigation provides limited information in the field of reproduction.

We respectively disagree. To our knowledge this is the first transcription factor, expressed in a sub-set of uterine stromal cells, shown to play an essential role during uterine remodelling and regeneration. As we show in Figure 8 removal of Blimp1 activity at PP day 0 has a massive effect on the ability of the uterus to undergo involution (Panel 8b emphasises the profound morphological differences between wild type and Blimp1 deficient uterine tissues at PP day 7).

Reviewer #2 (Remarks to the Author):

We thank the Reviewer for re-reviewing our revised manuscript and making additional constructive suggestions. In revising the manuscript we have taken into consideration all of their comments, have comprehensively addressed their concerns and provided further information and additional data as requested.

As I mentioned in the reviewer's comment, the submitted manuscript has interesting observations and the authors suggested an attractive hypothesis on the phenotypes of Blimp1 deficient uteri. I absolutely agree with the authors on the fact that uterine tissue resolution and following repair do not occur in Blimp1 mutant mice including a new model of Blimp1 deficient condition. However, the authors did not provide direct evidence to support the main theme of the submitted manuscript because of a lack of mechanistic approaches to demonstrate the direct role of Blimp1 in this process at the molecular level even in the revised manuscript. Although the authors included new experiments with additional genetic models for new Figures, such as Figures 8 and 9, they provide descriptive information again without any mechanistic approaches.

As mentioned above, in the absence of any specific suggestions as to how to address Blimp1 mechanism of action we carried out ATAC-seq experiments. Our results comparing wild type versus Blimp1 deleted tissue were included in the original rebuttal. This ATAC-seq data shows that Blimp1 mutant stromal cells are defective in their response to PR and ER signalling. These findings will form the basis for future longer term experiments aimed to explore the mechanistic basis of Blimp1 in uterine repair and regeneration.

Furthermore, it is now confusing since Figure 9 provides evidence that Blimp1 positive cells are restricted to a subset of stromal cells and not transformed to be included in the newly formed epithelium. The authors suggested the indirect role of Blimp1 expressed only in the stroma for epithelial regeneration in Figure 9. However, the initially submitted manuscript focused on the direct role of Blimp1 in epithelial repopulation during the repair process.

We originally included data showing that Blimp1 expression was upregulated in the stromal cells on the denuded surface of the placental detachment sites. This is clearly shown in Figure 7a (in the revised version). Given the large body of literature (see the reviews by Spooner et al 2021 & Ang et al 2023, cited heavily in our paper) arguing for the presence of a stromal stem cell population that contributes epithelial descendants during repair of the uterine luminal layer we speculated (and **only** speculated) in the original Discussion that these Blimp1+ cells might potentially represent these putative stem cells. While performing the revisions, and in response to the original Reviewer comments, we decided to definitely test this idea using a genetic lineage tracing approach. The new data is unambiguous – the stromal cells transiently expressing Blimp1 do **not** directly contribute to the restored CK+ epithelial layer, but rather act in a cell non-autonomous fashion to promote re-epithelialization. In the revised manuscript we present and discuss these new findings in the context of the current literature.

Although the phenotypes of Blimp1 deficient females are quite interesting and the authors suggested an interesting hypothesis, the revised manuscript does not contain data to demonstrate the direct role of Blimp1 on these events at the molecular level.

In the absence of specific suggestions as to how best investigate Blimp1 mechanism of action we carried out ATAC-seq comparing wild type versus Blimp1 deleted tissue. These ATAC-seq data, included in the rebuttal, demonstrate that Blimp1 inactivation renders the stromal cells defective in their response to PR and ER signalling.

1. Author response to comments 1 and 2

Since a variety of immune cells are positive for ER and/or PR and they are necessary for tissue clearance and following repair processes, it is critical to examine how infiltrated immune cells participate in these events as mentioned by the reviewer (comments 1 and 2 of reviewer 2). However, the authors considered this comment as “not relevant to this study” and did not pursue both comments related to this issue in the revised manuscript.

We chose not to consider the involvement of immune cells in tissue clearance as this has been investigated before. Thus previous studies have clearly established a role for enhanced decidual and uterine macrophage and CD4+ T-cell numbers and inflammatory responses following RU486 administration (Li et al. “Mifepristone (RU486) inducing abortion in a mouse model by regulating Innate and adaptive immune responses” *Reproductive and Developmental Medicine* 10.4103/2096-2924.288021). However to comprehensively address the Reviewers comment we have now examined macrophage populations in Blimp1 mutant uteri versus wild type uteri 24 hours post RU486 administration. We found F4/80 staining reveals that the population of macrophages in the uterine stroma is similar between wild type and mutant samples, but that, as predicted, we detect significantly enhanced macrophage numbers in the deciduae of wild type RU486 treated females prior to decidual detachment and release, compared to the situation in the Blimp1 deficient deciduae where the decidual swellings are refractory to RU486 induced detachment. These data have now been included in a new Figure (Supp Figure 4 provides both staining data and quantitation), and described in the Results section together with the reference to previously published studies.

2. Author response to comment 3

I think that the authors responded well to this comment and revised the manuscript accordingly.

3. Author response to the comment 4

The authors brought another genetic approach to answer comment 4. I mentioned above, I have no doubt about the evidence that timely post-partum tissue repair does not occur in the conditions that Blimp1 is deleted (including a conditional deletion of Blimp1). However, new Figures included in the revised manuscript (Figures 8 and 9) fail to demonstrate any direct role of Blimp1 in this process. These immunostaining-based data provide indirect and descriptive information again. Furthermore, the authors insist that Mmp10 and Blimp1 are co-expressed in the PP day 2 uterus in Figure 7. However, the results of IHC for Mmp10 and Blimp1 are not good enough to support what the authors mentioned in the revised manuscript and the quality of images is not convincing.

The comment that our experiments “fail to demonstrate any direct role of Blimp1 in this process” seems over-stated since the profound phenotypes we describe in both the gestational and post-partum uterus are **solely** attributable to the loss of Blimp1.

We show in serial tissue sections via IHC that nuclear Blimp1 and extra-cellular Mmp10 proteins are expressed both expressed in the placental detachment sites at PP day 2. In the original Figure the Reviewer commented that they found the images to be insufficiently convincing. We agree that in low power images the staining for Mmp10 looks very broad, and hence could be interpreted as being non-specific. However in our opinion the data are extremely robust, and indeed show in other Figures that the staining by both antibodies **is highly specific** – nuclear Blimp1 expression is lost in CreERT2 deleted samples (Supp Figure 4) as is Mmp10 in Blimp1 deleted samples (Figure 8). However to alleviate the Reviewers concern we have now provided high magnification images of Mmp10 and Blimp1 staining from the same regions of serial

sectioned samples. These images show that Mmp10 (a cytoplasmic and extra-cellular protein) is very robustly detected in the detachment sites but largely absent in the adjacent stroma. We show a very similar pattern of expression of nuclear Blimp1 staining, supporting our conclusion that both proteins are co-expressed in the stroma at detachment sites. We have tried repeatedly over the past 8 weeks to perform double IF on frozen sections. However due to the limitations of the commercially available Mmp10 antibodies we couldn't get the Mmp10 staining to work. We also attempted double staining for both antibodies using different fluorescently labelled secondary antibodies on the same wax sections (following antigen retrieval) to show that the expression patterns overlap in the detachment sites. Unfortunately the antigen retrieval conditions for the antibodies differ significantly and we were unable to find conditions that worked for both on the same sections. We hope that the improved higher magnification images now convincingly show that both Blimp1 and Mmp10 are highly up-regulated at the sites of placental detachment undergoing repair.

4. Author response to the comment 5.

It is quite understandable to have technical difficulties in performing co-immunofluorescence since available antibodies are limited. Although the authors mentioned that they performed CK and Blimp1 immunostaining in serial sections to demonstrate their colocalization, the revised manuscript does not have images to meet the comment. The authors also mentioned that Figure 9 with the fate mapping experiment could answer this comment. However, I am not sure how Figure 9 provided data to respond to comment 5. Figure 9 describes that Blimp1 expression is restricted to a subset of stromal cells underlying luminal epithelium, but does not give any further information.

Our original IHC staining experiments clearly show on serial sections that the Blimp1 expressing cells are closely intermingled with CK positive cells (Figure 7a). The Reviewer originally queried these findings and requested that we perform IF to show co-expression. In our rebuttal we explained that we had attempted this but that due to the very abundant levels of CK the IF experiment was not technically possible. In spite of acknowledging that it is "quite understandable to have technical difficulties in performing co-immunofluorescence since available antibodies are limited" the Reviewer states that "the revised manuscript does not have images to meet the comment".

Because the double labelling experiments the Reviewer requested were a) technically impossible and b) would only provide a static snap-shot of this population, we performed genetic lineage tracing experiments to rigorously test whether this superficial population of cells, which appear to co-express Blimp1 and CK, ultimately contribute to the newly formed epithelium. The results shown in Figure 9 provide cell fate mapping information tracking the cellular descendants of the Blimp1+ cells present in the surface layer of the denuded placental detachment sites. By pulse-labelling them using two independent Cre activatable reporter alleles (GFP and LacZ) to trace the descendants of this transient Blimp1+ population we unambiguously show they do not ultimately contribute to the newly formed luminal epithelium. In the case of the GFP lineage tracing experiments we now provide new images showing sections that were co-stained for Blimp1 and GFP staining. The merged images reinforce our conclusions that the Blimp1+/GFP double positive cells (yellow) give rise to green GFP positive progeny which are exclusively confined to the stromal cell population (Figure 9). We conclude that the superficial Blimp1+ cell population present at the denuded placental detachment sites do **not** represent a stem cell population previously reported by others as having the potential to contribute to epithelial repair. Rather we show that these Blimp1 expressing cells act non-cell autonomously to promote re-epithelialization and restoration of the luminal epithelial barrier.

5. Author response to the comment 6

Although the embryo may not be included in RU-486 treated group due to the reason that the authors suggested, it is doable to have a histologically matched area between the control and RU-486 treated groups as suggested by the reviewer. However, the authors did not pursue any further experiments to show better images to meet the comments in the revised manuscript.

We previously argued that the RU486 treated decidual section shown in Figure 2b was perfectly matched to the control un-injected sample as judged by the appearance of the placenta and the presence of the central maternal artery corresponding to the mid-point of the respective deciduae (readily visible in both untreated and treated sections). Moreover the point of the Panel was to compare the Blimp1 expression patterns in the decidual tissue not the embryo. Providing alternative sections by unnecessarily repeating experiments violates the UK Home Office guidelines that mandate that we minimise the number of animals used in experiments. However because the Reviewer is insistent we provide better images we have now obtained the necessary permission, and have collected additional RU486 treated samples that retained the deciduae within the uterine tissue to prevent collapse and extrusion of the dying embryo. We have replaced the panel in Figure 2 to show an image that includes a section through the embryo, and which shows the identical pattern of Blimp1 expression exclusively in the periphery of the deciduum as the original section.

6. Author response to the comment 7

I think that the authors responded well to this comment.